# AIRQA: A COMPREHENSIVE QA DATASET FOR AI RESEARCH WITH INSTANCE-LEVEL EVALUATION

**Tiancheng Huang**[1]**, Ruisheng Cao**[1]**, Yuxin Zhang**[1]**, Zhangyi Kang**[1]**, Zijian Wang**[1]**,**
**Chenrun Wang**[1]**, Yijie Luo**[1]**, Hang Zheng**[1]**, Lirong Qian**[1]**, Lu Chen**[1,2,3,4][†]**, Kai Yu**[1,3,4][†]

[1]X-LANCE Lab, School of Computer Science, Shanghai Jiao Tong University, Shanghai, China
[2]Shanghai Innovation Institution, Shanghai, China
[3]Jiangsu Key Lab of Language Computing, Suzhou, China
[4]Suzhou Laboratory, Suzhou, China
{htc981,chenlusz,kai.yu}@sjtu.edu.cn

## ABSTRACT

The growing volume of academic papers has made it increasingly difficult for researchers to efficiently extract key information. While large language models (LLMs) based agents are capable of automating question answering (QA) workflows for scientific papers, there still lacks a comprehensive and realistic benchmark to evaluate their capabilities. Moreover, training an interactive agent for this task is hindered by the shortage of high-quality interaction trajectories. In this work, we propose AirQA, a human-annotated comprehensive paper QA dataset in the field of artificial intelligence, with 13,956 papers and 1,246 questions, that encompasses multi-task, multi-modal and instance-level evaluation. Furthermore, we propose EXTRACTOR, an automated framework for instruction data synthesis. With three LLM-based agents, EXTRACTOR can perform example generation and trajectory collection without human intervention. Evaluations of multiple open-source and proprietary models show that most models underperform on AirQA, demonstrating its quality. Extensive experiments confirm that EXTRACTOR consistently improves the multi-turn tool-use capability of small models, enabling them to achieve performance comparable to larger ones.

## 1 INTRODUCTION

With the explosion of artificial intelligence (AI) publications, researchers must spend a significant amount of time reading lengthy papers just to locate a highly specific piece of information, which is both tedious and inefficient. The advent of large language models (LLMs), especially their remarkable reasoning and planning capabilities (Ahn et al., 2024; Guo et al., 2025; Huang et al., 2024; Wang et al., 2023b), has made it possible to automate the workflow of precise retrieval and question answering (QA) for academic papers (He et al., 2025; Othman, 2025; Skarlinski et al., 2024). Despite recent advances, there remains a notable absence of a comprehensive and realistic benchmark, which covers diverse question types and multi-modal abilities. And training an interactive QA agent that focuses on such task is difficult due to the scarcity of high-quality domain-specific trajectories.

Previous QA datasets on scientific papers usually focus on one narrow question type, such as querying technical details about a single paper (Dasigi et al., 2021; Lee et al., 2023; Pramanick et al., 2025; Singh et al., 2024), questions spanning across multiple documents following a rule-constructed two-hop pattern (Li et al., 2024), or aiming at the common paper retrieval requirements (Ajith et al., 2024; He et al., 2025). Accordingly, the evaluation function is usually tailored for one restricted type and lacks generalization to others. For example, M3SciQA (Li et al., 2024) designed one LLM-based prompt for long-form string evaluation with the reference answer, which is highly empirical and only serves its specific question type. On the other hand, most benchmark owners overly pre-process raw papers, and merely provide the cleaned text format for uniform input. This common

---

[*]Our code and data are publicly available at https://github.com/OpenDFM/AirQA.
[†]The corresponding authors are Lu Chen and Kai Yu.

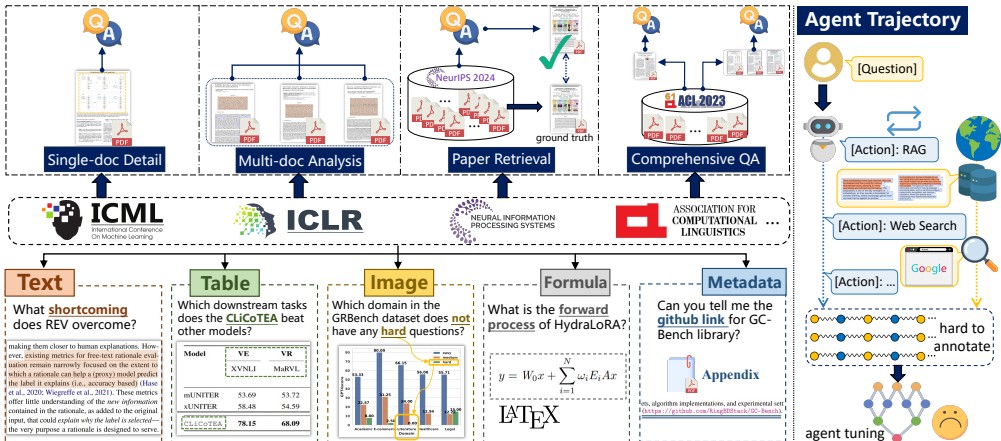

Figure 1: Left: An overview of the four question types and five element categories in our AirQA dataset. Right: An illustration of the bottleneck in multi-turn tool-use trajectory collection.

practice deviates from realistic scenarios, where real-world users may query other hyper-textual elements (illustrated in the bottom part of Figure 1) embedded in the raw PDF documents, such as figures, tables, formula, metadata, or even different combinations of them.

When tackling QA on academic papers, trivial methods (e.g. provide titles and abstracts alongside the question (Dasigi et al., 2021; Singh et al., 2024)) will easily fail due to context limitation, as the scaling of papers augment from a single paper to the entire conference volume. More advanced approaches adopt the popular RAG framework (Guu et al., 2020; Borgeaud et al., 2022; Izacard et al., 2022), but are not applicable when multi-turn reasoning over various chunked snippets are required. Meanwhile, agents which can predict executable retrievals or function-calling actions and interact with the outer environment for external knowledge exhibit significant potential in handling long-context multi-hop scenarios, making it a good choice for scientific QA under realistic and complicated settings (He et al., 2025; Nakano et al., 2022; Schick et al., 2023). Unfortunately, manually annotating task-specific trajectories of interactions with the environment is both time-consuming and expensive, requiring domain expertise, while simple data generation with LLMs can't faithfully synthesize $(action, observation)$ sequences with internal coherence and dependencies. As a result, the paucity of high-quality trajectory prevents the post-training of an effective QA agent.

To resolve the aforementioned bottlenecks, we propose a human-annotated multi-modal multi-task **A**rtificial **I**ntelligence **R**esearch **Q**uestion **A**nswering dataset, AirQA, which encompasses 1,246 examples and 13,956 papers in the domain of artificial intelligence, aiming at evaluating research capabilities in realistic scenarios. As illustrated in Figure 1, our dataset contains 4 different question types and 5 different element categories, with 19 parameterized Python functions to support customized evaluation. Furthermore, to advocate agentic model post-training, we propose a multi-agent framework, ExTRACTOR, for instruction data synthesis, which includes an explorer that generates natural language QA pairs based on contexts from papers, a tracker that rewrites QA pairs into properly formatted examples, and an actor that interacts with the environment to collect trajectories.

We evaluate a wide range of open-source and proprietary LLMs on different baselines. Performances show that, though given several external information sources, LLMs struggle on our AirQA dataset, with the best model scoring only 44.14% overall, indicating that existing workflows are still underdeveloped. With the proposed ExTRACTOR framework, we fine-tune models of different sizes. Results show that, with just 4,000 interaction trajectories, fine-tuned 7B model achieves a performance comparable to untrained 14B model. Extensive experiments demonstrate that, the accuracy raises consistently as data scales up, highlighting the scalability of our framework.

To summarize, our contributions are threefold:

- We propose AirQA, a human-annotated multi-modal multi-task multi-paper QA dataset with function-based instance-specific evaluations. To the best of our knowledge, AirQA is the first dataset that encompasses multiple question types, also the first to bring function-based evaluation into QA domain, enabling convenient and systematic assessment of research capabilities.

- We introduce EXTRACTOR, a document-based framework aiming at the synthesis of QA examples, interaction trajectories and instruction data, serving as an empirical method for improving the agent's multi-turn tool-using ability without the involvement of manual annotation.
- We evaluate various LLMs and different QA baselines on our AirQA dataset, demonstrating the quality of our dataset, and indicating the insufficiency of current methods. Extensive experiments on instruction tuning reveal that, small models significantly benefit from our synthetic instruction data, validating the effectiveness of our proposed EXTRACTOR framework.

## 2 THE AirQA DATASET

In this section, we introduce the task definition, the evaluation metrics, the construction and the statistics of our AirQA dataset.

### 2.1 TASK DEFINITION

To more effectively evaluate existing models and methods across a broader range of tasks, rather than limiting assessment to individual tasks, we carefully analyze real-world AI research scenarios, and systematically design the following four question types in AirQA to cover them up:

Table 1: Examples of different question types from our AirQA dataset.

| Type | Question | Answer Format |
|---|---|---|
| single | Which downstream tasks does the CLiCoTEA outperform other models in terms of zero-shot performance on the IGLUE benchmark? | Your answer should be a Python list of strings, every string is the abbreviation of a downstream task type mentioned in the paper. |
| multiple | According to this survey, what're the three most recent decoder-only LLMs for NL2Code? How many programming languages do their training datasets each contain? | Your answer should be a Python dictionary of 3 key-value pairs, where each key is a string, the LLM, and each value is the number of programming languages. |
| retrieval | Which paper unifies reinforcement learning and imitation learning methods under a dual framework? | Your answer should be the exact title of the paper WITHOUT ANY OTHER EXPLANATION. |
| comprehensive | Among the text-to-SQL papers in ACL 2023, which one achieves the best testsuite accuracy on the SPIDER dataset? Tell me the paper title and corresponding test accuracy. | Your answer should be a Python list of length two, with the first one being the title string and the second one being a float, the accuracy rounded to 3 decimals. |

**Single-doc Detail** Querying detailed information from a specific paper. Besides text, we also explore different textual and non-textual aspects including table, image, formula and metadata to cover all elements that may appear in a scientific paper. We showcase one example for each category in Figure 1. Notably, a question may belong to multiple categories, requiring diverse capabilities.

**Multiple-doc Analysis** Posing questions across multiple papers. A simple idea for constructing multiple-doc questions is to merely combine several single-doc questions, but it overlooks the possible relations between different papers, which are actually what researchers pay more attention to. To imitate the real scenes where researchers scan across several documents to find the answer to a question, we propose two paradigms: 1) compare same aspects of different papers, and 2) find subtle points that are not fully illustrated in one paper, and explore the details in the papers it cites.

**Paper Retrieval** Retrieving papers from a specific conference in a particular year, based on the description. Considering the search scale, while Skarlinski et al. (2024) argues that retrieval on a fixed corpus is not suitable as performance proxies for real scientific research tasks, we insist that a dataset cannot contain an infinite number of papers. Without limitation, the answer would be ambitious, making the evaluation unfair. Only retrieval on a fixed corpus can ensure the objectivity of the dataset. Among these questions, 240 are directly transformed from author-written questions in the LitSearch (Ajith et al., 2024) dataset with rule-based conversion.

**Comprehensive QA** A combination of the three aforementioned question types. Specifically, a comprehensive QA question may combine a retrieval question with either a single question or a multiple question. As an integrated task, this combination is designed for scenarios in which the

user cannot directly provide the paper or has forgotten the specific paper to which the question refers, but recalls certain key points, thereby enabling retrieval. The solution can be divided into two main stages: retrieving the paper based on its description, and answering the detailed question.

We also exhibit one example for each question type in Table 1. For brevity, in the following sections, we refer to the four question types as single, multiple, retrieval and comprehensive respectively. For retrieval and comprehensive questions, to ensure uniqueness of the retrieved paper, and to avoid ambiguity, we limit the scope of retrieval to be one of ACL2023, ICLR2024 and NeurIPS2024.

## 2.2 EVALUATION METHOD

As mentioned in Section 5, most previous QA datasets depend on linguistics metrics and LLMs for assessment, favoring semantic coherence over factual correctness, which holds little value under current circumstances. While in our AirQA dataset, we mainly focus on judging the correctness of the answer as objective as possible. We notice that, though the answers to different questions vary, they share common features. For example, when answering questions related to quantitative comparison, we only care about the number itself, rather than whether LLMs form a complete sentence. In this case, the number is the "scoring point" of this question, which directly determines the quality of the answer. Inspired by the instruction following ability of LLMs, we adopt output reformatting by providing an answer format along with the question (as shown in Table 1), such as, "*Your answer should be a Python list of two floats, each rounded to 2 decimal places.*". Our empirical check on 30 randomly sampled answers from Qwen2.5-3B-Instruct (Qwen et al., 2025) further shows that the model adheres to the answer format in over 90% of cases (28/30), indicating that output reformatting is quite applicable in practice even for smaller models. In this way, we guide LLMs to output the scoring points we primarily concern with, benefiting the following evaluation.

To evaluate scoring points of different kinds, we design 19 Python functions and complement them with optional keyword arguments (e.g. `ignore_order` for list comparison) to support example-specific assessment. For each evaluation, the final result will be either 0 or 1, representing wrong and right. Based on whether they utilize LLMs for semantic judgment or not, and their functionalities, these functions can be classified into two types and six subtypes as shown in Table 8. An evaluation function is subjective if it involves LLMs, and is objective if not. Specifically, for logical functions, which combine multiple functions in one evaluation, the evaluation is classified as subjective as long as there is one subjective function. For subjective functions, we select GPT-4o-mini-2024-07-18 as the backbone model for its relative stability. More details of the functions can be found in App. A.3.

To further clarify the role and reliability of subjective evaluation, we highlight three points: 1) While LLM-based judgment is necessary in certain cases, we design tailored prompts to support more fine-grained and targeted evaluation unlike previous datasets which mostly rely on a fixed prompt. e.g., We design a specialized prompt to compare LaTeX formulas. 2) Existing studies (Wang et al., 2023a; Ho et al., 2025; Kamalloo et al., 2024) also suggest that, for QA tasks, LLM-based evaluations are more aligned with human judgments than metrics such as accuracy or F1. 3) Analysis on 66 examples shows that LLM-based and human evaluations are largely consistent, with an agreement rate of approximately 83%.

## 2.3 DATASET CONSTRUCTION

**Annotators** To ensure professionalism, we employ 26 students with expertise in artificial intelligence. Their task is threefold: 1) read a paper they are interested in, 2) pose an answerable question based on the textual and non-textual content of the paper (and additional papers if needed) they read, in accordance with the aforementioned question types, 3) wrap the question, the evaluation function, and other necessary information into an example file, as presented in App. A.1. Example files are then sent into an automated inspection pipeline, and annotators are asked to rewrite unqualified ones.

**Paper Collection** Due to the professional background of the annotators, all papers are selected from the field of artificial intelligence to ensure accurate comprehension of the content. To facilitate reproduction, we assign an uuid for each used paper based on its title and its conference. We also generate a metadata file for each paper, containing the title, the abstract, the URL where the paper is downloaded, and other information. For further illustration, please refer to App. A.2.

## 2.4 DATASET STATISTICS

**Example Classification**   We classify the examples in the AirQA dataset into four question types, five element categories and two evaluation types as discussed before. Table 2 shows that, the example numbers of the four question types are relatively balanced, while approximately half of the examples involve at least one element other than text (we classify an example as text if and only if it doesn't include any other elements, and an example can belong to more than one categories). Regarding evaluation functions, Figure 2 shows that most of the evaluations are objective, meaning they do not require LLMs, which demonstrates the cost-effectiveness of our dataset.

**Paper Usage**   To ensure the objectiveness of retrieval and comprehensive questions, we limit the retrieval scale to be one of ACL2023, ICLR2024 and NeurIPS2024. Besides including all papers from these three conferences in our collection, we also utilize another 707 papers in the examples, summing up a total of 13,956 papers. As shown in Table 2, in average, an example involve 1.63 papers, indicating the diversity of our dataset.

Table 2: Statistics of examples. For the last two statistics, we only consider single and multiple questions.

| Statistics | Number |
|---|---|
| **Question Type** | |
| - single | 351(28%) |
| - multiple | 323(26%) |
| - retrieval | 288(23%) |
| - comprehensive | 284(23%) |
| **Element Category** | |
| - text | 621(50%) |
| - table | 213(17%) |
| - image | 207(17%) |
| - formula | 127(10%) |
| - metadata | 122(10%) |
| **Overall** | **1246(100%)** |
| Avg. question length | 34.52 |
| Max. question length | 118 |
| Avg. # papers per example | 1.63 |
| Max. # papers per example | 7 |

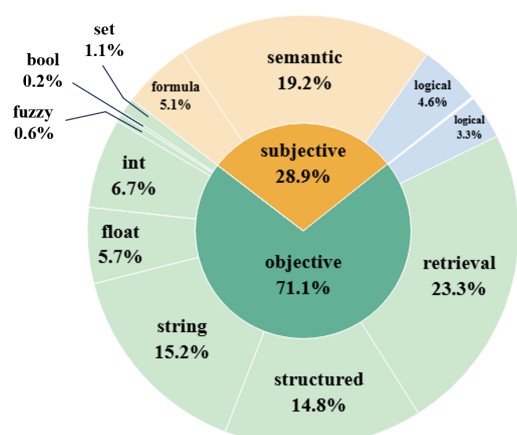

Figure 2: Distribution of different evaluation categories. 'bool', 'int', 'string', 'fuzzy', 'structured' stand for specific evaluation functions in 'match' subtype.

**Comparison with Existing Datasets**   In Table 3, we compare AirQA with existing scientific QA datasets. It is evident that AirQA demonstrates several salient strengths: 1) **More question types.** AirQA designs four different question types to systematically cover realistic research scenarios, 2) **More element types.** AirQA contains a wider variety of elements, including table, image, formula and metadata, 3) **More precise evaluation.** AirQA employs 19 parameterized functions, which can be classified into two types and six subtypes, facilitating customized evaluation.

Table 3: Comparison of our AirQA dataset and existing scientific QA datasets.

| Dataset | # QA | Evaluation Methods | Task types | | | | Question based on | | | | |
|---|---|---|---|---|---|---|---|---|---|---|---|
| | | | Sgl. | Multi. | Retr. | Comp. | Full Text | Table | Image | Form. | Meta. |
| ScholarlyRead (Saikh et al., 2020) | 10K | BLEU, METEOR, ROUGE | ✓ | ✗ | ✗ | ✗ | ✗ | ✗ | ✗ | ✗ | ✗ |
| QASPER (Dasigi et al., 2021) | 5,049 | F1 | ✓ | ✗ | ✗ | ✗ | ✗ | ✗ | ✗ | ✗ | ✗ |
| QASA (Lee et al., 2023) | 1,798 | Precision, Recall, F1, ROUGE | ✓ | ✗ | ✗ | ✗ | ✓ | ✗ | ✗ | ✗ | ✗ |
| SPIQA (Pramanick et al., 2025) | 270K | METEOR, CIDEr, ROUGE, BERTScore, LLMLogScore | ✓ | ✗ | ✗ | ✗ | ✓ | ✓ | ✓ | ✗ | ✗ |
| PeerQA (Baumgärtner et al., 2025) | 579 | MRR, Recall, Rouge-L, AlignScore, Prometheus-2 | ✓ | ✗ | ✗ | ✗ | ✓ | ✗ | ✗ | ✓ | ✗ |
| SciDQA (Singh et al., 2024) | 2,937 | ROUGE, BLEURT-20, BERTScore, LLM judge | ✓ | ✓ | ✗ | ✗ | ✓ | ✓ | ✓ | ✓ | ✗ |
| M3SciQA (Li et al., 2024) | 1,452 | MRR, LLM judge | ✗ | ✓ | ✗ | ✗ | ✓ | ✓ | ✓ | ✗ | ✗ |
| AutoScholarQuery (He et al., 2025) | 35K | Precision, Recall | ✗ | ✗ | ✓ | ✗ | ✓ | ✗ | ✗ | ✗ | ✗ |
| LitSearch (Ajith et al., 2024) | 597 | Recall | ✗ | ✗ | ✓ | ✗ | ✓ | ✗ | ✗ | ✗ | ✗ |
| LitQA2 (Skarlinski et al., 2024) | 248 | Precision, Accuracy | ✗ | ✗ | ✗ | ✓ | ✓ | ✗ | ✗ | ✗ | ✗ |
| **AirQA (Ours)** | 1,246 | Instance-level Function | ✓ | ✓ | ✓ | ✓ | ✓ | ✓ | ✓ | ✓ | ✓ |

# 3   EXTRACTOR FRAMEWORK FOR TRAJECTORY SYNTHESIS

In this section, we introduce our trajectory synthesis framework, EXTRACTOR, based on its three components: explorer, tracker and actor.

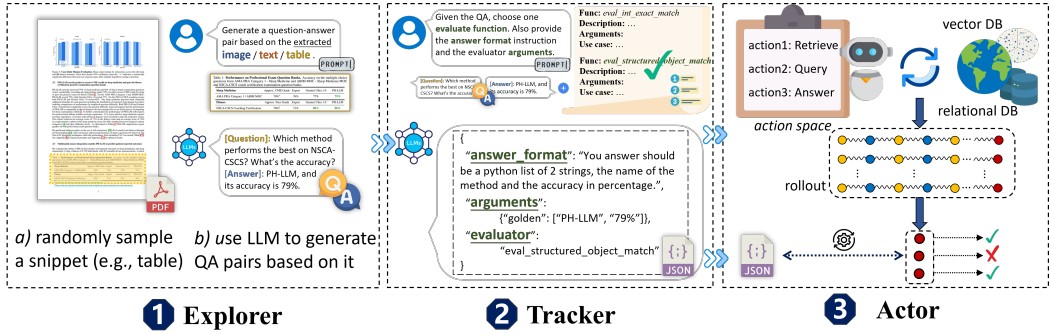

Figure 3: An overview of our EXTRACTOR framework, which consists of three stages to automatically extract QA pairs, formulate evaluation and answers, and filter valid agent trajectories.

## 3.1   OVERALL FRAMEWORK

To handle scientific QA, there are mainly three types of methods: 1) offering relevant information of the paper (e.g., title and abstract) alongside the question, 2) providing additional contexts via retrieval methods (e.g., RAG), and 3) equipping LLMs with supplementary tools, so that they can obtain sufficient information during multi-turn interactions. As discussed in Section 4.2, methods of the former two types significantly underperform the latter, even when supported by superior backbone models. Therefore, for the following fine-tuning, we apply Agentic Hybrid baseline (further illustrated in Section 4.1), whose environment includes a database and a vectorstore that produce execution or query results as observations when called by the agent with two predefined actions.

To mimic the real-world annotation and interaction scenarios, we split the synthesis process into three separate stages: 1) exploration stage, constructing a natural language question-answer pair with given context, 2) tracking stage, choosing suitable evaluation function and fill in the formatted example file, and 3) action stage, interacting with the outer environment to collect trajectories.

## 3.2   EXPLORER

Above all, we randomly download 10,000 papers in the artificial intelligence domain from `arXiv`, collect their metadata including titles and abstracts, and employ PyMuPDF (Artifex Software, 2023) and `MinerU` (Wang et al., 2024) to extract both textual and non-textual elements from the papers.

For the explorer, its goal is to generate rational question-answer pairs with sampled contexts. Based on different question types, we design three different modes: 1) For single type, we first randomly choose a paper and an element. Then, corresponding contexts are extracted according to the category of the element, and the explorer is expected to output a long-form QA pair. 2) Regarding retrieval type, instead of contexts, the explorer only receives the title and abstract as inputs, and is required to generate a QA pair in which the answer is the paper title and the question is designed to pinpoint the corresponding paper. 3) As for comprehensive type, we basically follow single type, while the only difference is that we provide the title and abstract along with the context, and ask the explorer to elicit the paper where the element comes from in the question. To improve the quality of QA pairs, we adopt chain-of-thought (Wei et al., 2023) and hand-written category-based hint prompts.

## 3.3   TRACKER

Regarding the tracker, its purpose is to wrap the previously generated natural language QA pairs into example files in accordance with specific formats. As explorer settings for different question types vary, we employ different tracker settings: 1) With regard to single and comprehensive questions,

we provide the tracker with the QA pair along with the information of the evaluation functions, including descriptions, parameters and use cases. The tracker is then asked to choose the suitable evaluation function, fill in the parameters and the answer format, and refine the QA pair accordingly. 2) In terms of retrieval questions, as we restrict the answer to be the exact title of the chosen paper, we simply fill in the example file with fixed evaluation function, parameters and answer format. 3) As for multiple questions, the manual annotation consistently involves new papers outside our sampled collection, requiring real-time download and processing, which is incompatible with our explorer agent. For simplicity, we propose a rule-based combination of single examples: a) merging questions and answer formats with natural language templates, and b) combining evaluation functions with the logical function `evaluate_conjunction` (function details in App. A.3).

For more specific explorer and tracker prompts, please refer to App. E.

### 3.4 ACTOR

As for the actor, it aims at interacting with the environment to collect trajectories for instruction construction. In the outer environment, we include a database and a vectorstore containing corresponding information of the papers. Following Zeng et al. (2023), we employ ReAct (Yao et al., 2023) framework with three actions to interact with the environment. For each synthetic example, we use LLM as an actor to produce an interaction trajectory in a message list manner $(u_0, a_0, \ldots, u_i, a_i, \ldots, u_n, a_n)$, where $u_i$ represents the user's instruction, or the observation from the environment, and $a_i$ denotes the response from the actor, including a thought and an action.

To avoid exceeding context length, we adopt the idea of sliding window and chunk the message list based on a window size of 5, generating multiple instruction data from one trajectory. During training, for each chunked list, we mask previous message history and train the last turn only.

We also observe that, some errors appear frequently in the collected trajectories (e.g., attempts to utilize undefined parameters). To ensure data quality, we remove instruction data that ends with a wrong action. For other instruction data, we reserve previous wrong actions and corresponding error information in the message list to guarantee error correction capability and coherence of thoughts.

## 4 EXPERIMENT

### 4.1 EXPERIMENT SETTINGS

**Baselines** To comprehensively assess the LLMs on AirQA, we implement 8 baselines, including:

- **Trivial Baselines**: 1) Question Only baseline, only the question and the corresponding answer format are available, 2) Title-Abstract baseline, the titles and the abstracts of the corresponding papers are provided alongside, and 3) Full-Text with Cutoff baseline, raw textual contexts extracted from the papers are given in limited length.

- **Retrieval Baselines**: 1) RAG baseline, the question is sent to the vectorstore to retrieve relevant chunks, and LLMs answer the question based on retrieved contexts, and 2) Text2SQL baseline, where LLMs first generate a SQL, then answer the question based on the query results.

- **Agentic Baselines**: We employ ReAct (Yao et al., 2023) framework with three actions: RE-TRIEVE, QUERY and ANSWER, corresponding to retrieving from the vectorstore, querying the database and generating the final answer. With this framework, we implement 1) Agentic RAG and 2) Agentic Text2SQL baseline that only interact with the vectorstore and the database, respectively. 3) Agentic Hybrid baseline with all actions. Details on actions can be found in App. B.

Note that for both base and fine-tuned models, all evaluations are performed on AirQA to enable a clear and fair comparison. Examples generated by EXTRACTOR are for training only.

**LLMs and Hyper-Parameters** We evaluate various LLMs on AirQA. For closed-source ones, we use GPT-4o-2024-08-06, o1-mini-2024-09-12, Claude-3.7-Sonnet-20250219 and Gemini-2.5-Pro-exp-03-25. Regarding open-source LLMs, we employ Qwen2.5-72B-Instruct (Qwen et al., 2025), Llama-3.3-70B-Instruct (Hugging Face Team, 2023), and DeepSeek-R1 (Guo et al., 2025). As for hyper-parameters, the `temperature` is set to 0.7 and `top_p` is fixed to 0.95. Specifically, for

reasoning models, the `temperature` is set to $0.6$. The maximum retrieved tokens in each turn and the cutoff for full-text input are both limited to 5K. The threshold of interaction turns is 20 and the window size for the message history is 5. For closed-source models, we directly call their API services, while for open-source ones, we deploy them on NVIDIA A800 Tensor Core clusters using vLLM [1] (Kwon et al., 2023).

**Instruction Tuning** For instruction tuning, we choose the Qwen2.5 family (Qwen et al., 2025) as base models. For all three agents in our EXTRACTOR framework, we employ Qwen2.5-32B-Instruct. While for the target model, unless otherwise specified, we utilize Qwen2.5-7B-Instruct as the backbone. As for the training framework, we employ LLaMA-Factory (Zheng et al., 2024). Regarding our synthetic instruction data, we transform them into standardized ShareGPT format following Vicuna (Chiang et al., 2023), and as mentioned before, we only compute the loss of the model's last output during fine-tuning by setting `mask_history` as true. By default, we use a learning rate of $1 \times 10^{-4}$, apply AdamW optimizer (Loshchilov & Hutter, 2019) with a cosine learning scheduler and train for one epoch. The entire training process is conducted on two Ascend 910B4 NPUs with 64GB of memory each.

The detailed hyper-parameters used in LLaMA-Factory for instruction tuning are listed in Table 9.

## 4.2 MAIN RESULTS

**Evaluation of Base Models** To figure out different baselines' performance on the AirQA dataset, we choose two widely used models, GPT-4o and Qwen2.5-72B-Instruct, as representatives of proprietary and open-source LLMs. Table 4 shows that: 1) **Trivial baselines perform poorly.** Under Question Only setting, LLMs can only answer $5\%$ of the questions correctly, demonstrating the quality of our AirQA dataset. 2) **Provided more information sources, LLMs produce better answers.** With just a glimpse into the database or the vectorstore, retrieval baselines elevate the overall accuracy by at least $8\%$ compared to Question Only baseline. 3) **As the interaction turn increases, LLMs explore the backend environment better.** While both agentic baselines outperform their retrieval counterparts, Agentic Text2SQL baseline exhibits significantly greater improvement, indicating that for structured retrieval, more searches enable step-by-step problem-solving, while for unstructured retrieval regarding the vectorstore, a single query is sufficient for most circumstances.

Table 4: Performance of different baselines on AirQA.

| Baseline | Question Type | | | | Element Category | | | | | Evaluation | | AVG |
|---|---|---|---|---|---|---|---|---|---|---|---|---|
| | sgl. | multi. | retr. | comp. | text | table | image | form. | meta. | obj. | subj. | |
| GPT-4o-2024-08-06 | | | | | | | | | | | | |
| Question Only | 8.55 | 1.86 | 1.04 | 5.63 | 4.35 | 1.41 | 10.63 | 2.36 | 0.00 | 3.95 | 5.54 | 4.41 |
| Title-Abstract | 11.40 | 5.26 | 0.00 | 5.28 | 5.96 | 4.23 | 8.70 | 4.72 | 2.46 | 4.07 | 9.97 | 5.78 |
| Full-Text w/ Cutoff | 33.90 | 8.05 | 0.69 | 5.99 | 13.53 | 7.51 | 13.53 | 12.60 | 18.03 | 9.94 | 21.05 | 13.16 |
| RAG | 31.62 | 4.95 | 18.75 | 16.55 | 20.29 | 12.68 | 16.91 | 17.32 | 18.03 | 18.19 | 18.56 | 18.30 |
| Text2SQL | 21.08 | 6.81 | 7.64 | 17.25 | 14.01 | 8.92 | 12.08 | 16.54 | 14.75 | 11.41 | 18.28 | 13.40 |
| Agentic RAG | 34.19 | 8.36 | 15.63 | 29.58 | 21.36 | 18.78 | 26.57 | 22.83 | 24.59 | 21.36 | 24.10 | 22.15 |
| Agentic Text2SQL | 42.17 | **11.15** | 18.40 | **38.38** | 23.19 | 21.60 | **28.99** | 33.07 | **47.54** | 26.44 | **31.02** | 27.77 |
| Agentic Hybrid | **45.58** | 10.53 | **52.13** | 35.56 | **39.61** | **23.00** | 25.60 | **33.86** | 47.54 | **38.76** | 29.09 | **35.96** |
| Qwen2.5-72B-Instruct | | | | | | | | | | | | |
| Question Only | 9.69 | 1.86 | 0.35 | 5.99 | 2.74 | 3.29 | 10.63 | 3.94 | 5.74 | 4.52 | 4.99 | 4.65 |
| Title-Abstract | 17.66 | 6.19 | 0.00 | 8.10 | 8.05 | 6.10 | 12.08 | 7.87 | 7.38 | 6.44 | 13.30 | 8.43 |
| Full-Text w/ Cutoff | 36.18 | 8.98 | 0.00 | 7.04 | 12.56 | 11.27 | 16.91 | 14.17 | 18.85 | 11.86 | 19.67 | 14.13 |
| RAG | 31.91 | 7.43 | 18.75 | 21.83 | 22.06 | 11.27 | 19.32 | 20.47 | 21.31 | 19.55 | 21.88 | 20.22 |
| Text2SQL | 22.22 | 4.02 | 11.11 | 13.38 | 13.85 | 8.45 | 15.46 | 10.24 | 11.48 | 12.43 | 14.13 | 12.92 |
| Agentic RAG | 32.76 | 9.60 | 15.63 | 30.28 | 22.06 | 15.96 | 25.12 | 25.98 | 18.85 | 21.02 | 25.21 | 22.23 |
| Agentic Text2SQL | **43.02** | **11.46** | 43.40 | **40.14** | 36.07 | **21.13** | **29.95** | **35.43** | 49.18 | 35.37 | **31.59** | 34.27 |
| Agentic Hybrid | 39.03 | 10.84 | **55.21** | 37.32 | **41.71** | 13.15 | 28.02 | 30.71 | 45.90 | **37.74** | 28.53 | **35.07** |

While baselines such as Question Only baseline exhibit a relatively low performance, Agentic Hybrid baseline consistently outperforms the others, allowing for comparisons between different models. In Table 5, we can observe that: 1) **Proprietary models outperform open-source ones,** show-

---

[1] https://docs.vllm.ai/en/latest/index.html

ing a stronger research capability, while some open-source models achieve performance comparable to closed-source ones. 2) **Reasoning models' performance on this method is not satisfactory,** possibly due to the incompatibility between their fixed reasoning formats and our framework.

Table 5: Performance of Agentic Hybrid baseline with different backbone models on AirQA.

| Model | Question Type | | | | Element Category | | | | | Evaluation | | AVG |
|---|---|---|---|---|---|---|---|---|---|---|---|---|
| | sgl. | multi. | retr. | comp. | text | table | image | form. | meta. | obj. | subj. | |
| GPT-4o | 45.58 | 10.53 | 52.13 | 35.56 | 39.61 | 23.00 | 25.60 | **33.86** | 47.54 | 38.76 | 29.09 | 35.96 |
| o1-mini | 37.04 | 12.07 | 45.14 | 24.65 | 35.43 | 14.55 | 22.22 | 22.83 | 36.07 | 31.07 | 26.04 | 29.61 |
| Claude-3.7-Sonnet | 45.30 | 15.17 | 58.68 | 27.46 | 43.96 | 22.07 | 24.64 | 27.56 | 44.26 | 39.32 | 29.64 | 36.52 |
| Gemini-2.5-Pro | **51.85** | **18.58** | **67.01** | **40.49** | **51.53** | **29.58** | **29.95** | 33.86 | **53.28** | **46.55** | **38.23** | **44.14** |
| Qwen2.5-72B-Instruct | 39.03 | 10.84 | **55.21** | **37.32** | **41.71** | 13.15 | **28.02** | **30.71** | **45.90** | **37.74** | **28.53** | **35.07** |
| Llama-3.3-70B-Instruct | 29.06 | 9.29 | 42.71 | 24.30 | 32.37 | 8.92 | 21.74 | 19.69 | 30.33 | 28.47 | 19.94 | 26.00 |
| DeepSeek-R1 | **41.03** | **11.46** | 41.67 | 22.54 | 35.10 | **15.96** | 20.77 | 20.47 | 39.34 | 30.40 | 26.59 | 29.29 |

**Evaluation of Fine-tuned Models**   We fine-tune three models, 3B, 7B and 14B, with instruction data extracted from the 4,000 trajectories. Table 6 shows that, all three exhibit improved performance after training. The observed reduction in the 14B model's performance gain is considered reasonable and acceptable, because: 1) The actor agent in our EXTRACTOR framework in effect serves as a teacher agent. The selected teacher model Qwen2.5-32B-Instruct scores only 31.94% on AirQA, which represents the upper performance bound achievable through distillation. 2) While this diminishing return has long been a meaningful and widely discussed research question beyond the scope of this paper, we can observe that with EXTRACTOR, small models produce significantly less errors in predicting actions. As shown in Table 7, 7B's error action rate drops from 38.69% to 6.85%, and similar improvement is observed on 14B, with error action rate dropping from 31.63% to 6.64%, indicating the effectiveness of our framework.

Table 6: Performance of models trained using EXTRACTOR and evaluated on AirQA. "FT" denotes fine-tuning.

| Size | FT? | Question Type | | | | Element Category | | | | | Evaluation | | AVG |
|---|---|---|---|---|---|---|---|---|---|---|---|---|---|
| | | sgl. | multi. | retr. | comp. | text | table | image | form. | meta. | obj. | subj. | |
| 3B | ✗ | 7.98 | 2.48 | 12.85 | 6.69 | 9.5 | 3.29 | 6.28 | 4.72 | 7.38 | 7.91 | 6.09 | 7.38 |
| | ✓ | 14.81 | 3.72 | 51.74 | 13.73 | 29.79 | 4.23 | 10.63 | 11.81 | 17.21 | 24.97 | 8.59 | 20.22 |
| 7B | ✗ | 16.24 | 3.72 | 26.39 | 15.85 | 19.48 | 8.45 | 13.53 | 4.72 | 14.75 | 17.29 | 10.25 | 15.24 |
| | ✓ | 21.08 | 4.95 | 51.04 | 22.18 | 33.66 | 7.51 | 14.01 | 13.39 | 25.41 | 27.01 | 16.90 | 24.07 |
| 14B | ✗ | 25.07 | 7.74 | 46.18 | 25.35 | 31.88 | 10.33 | 22.22 | 18.90 | 24.59 | 28.81 | 17.45 | 25.52 |
| | ✓ | 25.36 | 6.19 | 52.08 | 26.41 | 34.94 | 7.51 | 20.77 | 19.69 | 28.69 | 30.96 | 16.62 | 26.81 |
| 32B | ✗ | 36.47 | 11.76 | 52.78 | 28.17 | 38.81 | 13.62 | 24.15 | 26.77 | 37.7 | 34.8 | 24.93 | 31.94 |

## 4.3   ABLATION STUDY

**Component Ablation**   To evaluate the effect of the two proposed components, sliding window and error removal, we fine-tune two additional models: (1) one without either component, which uses raw trajectories as training data and computes loss over all model outputs, and (2) another that applies sliding window but retains all error actions. As shown in Table 7, sliding window yields a more pronounced improvement in the overall score, and a drastic reduction is also observed in error action rate, defined as $\frac{\#\ error\ actions}{\#\ actions}$, demonstrating the value of error removal in improving the model's ability to generate valid actions.

Table 7: Component Ablation.

| Setting | Overall (%) | Error Rate(%) |
|---|---|---|
| base model | 15.24 | 38.69 |
| EXTRACTOR
- w/o sliding window
- w/o error removal | 20.47 | 26.25 |
| EXTRACTOR
- w/o error removal | **24.08** | 20.32 |
| EXTRACTOR | 24.07 | **6.85** |

**Synthetic Data Scale**   We fine-tune another four models with more instruction data extracted from 1K, 2K, 4K and 10K trajectories. Figure 4 shows that, as the number of trajectory increase, the scores of all question types raise consistently, demonstrating the scalability of our method.

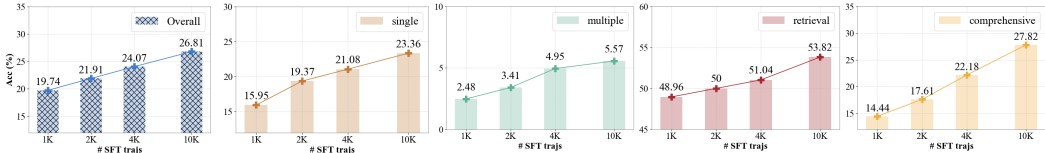

Figure 4: Performance changes with the increasing number of synthesized trajectories.

Due to page limit, more experimental results and analysis regarding AirQA and EXTRACTOR are presented in App. C and App. D.

## 5 RELATED WORK

**PDF-based Scientific QA Datasets** Previous research has consistently focused on the development of high-quality scientific QA datasets. Some datasets emphasize the accurate retrieval of papers based on specific descriptions (Ajith et al., 2024; He et al., 2025), while others concentrate on extracting detailed information from the text (Baumgärtner et al., 2025; Dasigi et al., 2021; Lee et al., 2023; Jin et al., 2019; Saikh et al., 2020). Recent work advance this field by incorporating non-textual elements (Pramanick et al., 2025; Singh et al., 2024). Li et al. (2024) further extend this approach by introducing a paradigm for generating cross-document questions, while Skarlinski et al. (2024) pay more attention to the combination of paper retrieval and detailed QA. Our AirQA dataset innovates in this field by encompassing various question types and element categories, while designing a function-based instance-level evaluation.

**Instruction Tuning and Synthetic Data** Instruction tuning serves as a useful tool for aligning LLMs with human instructions, but it requires corresponding outputs for specific instructions, thus heavily relying on high-quality training data. While manual annotation is an effective method, it is hard to scale up due to time and cost constraints. Wan et al. (2024), Wang et al. (2023c) and Chen et al. (2025) introduced different methods for crafting synthetic instruction data from scratch, while Zeng et al. (2023) proposed extracting data from interaction trajectories to leverage existing datasets. In this work, we introduce a multi-agent framework, EXTRACTOR, to automate both example generation and trajectory collection, facilitating instruction data synthesis.

## 6 CONCLUSION

In this work, we manually annotate a multi-modal multi-task multi-paper dataset (AirQA) with instance-level evaluation, and a multi-agent framework (EXTRACTOR) for instruction data synthesis. Evaluations demonstrate the quality of our dataset, indicating the challenges current models face in scientific QA, while experiments on instruction tuning highlight the effectiveness of the framework. Future works include: 1) exploring other RL-based methods for further improvements, and 2) extend the PDF-based tasks to other knowledge intensive domains, including law and medicine.

ACKNOWLEDGMENTS

We would like to thank Haoran Wang, Jingyi Zhang, Ye Wang, Yuxun Miao, Danyang Zhang, Hanqi Li, Zichen Zhu, Situo Zhang, Senyu Han, Dingye Liu, Wenjie Sun, Hanchong Zhang, Nan Jiang, Liangtai Sun, Da Ma, Hankun Wang, and Zhihan Li for their careful annotation on the AirQA dataset. This work is funded by the China NSFC Projects (62120106006, U23B2057, 92370206, 62576212) and Shanghai Municipal Science and Technology Project (25X010202846).

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

## ETHICS STATEMENT

To construct our AirQA dataset, we use 13,956 papers in artificial intelligence domain. Most papers utilized in our AirQA dataset can be downloaded from arXiv [2]. For some conference papers unavailable on arXiv, we use OpenReview [3] and ACL Anthology [4] as supplements. All three websites are licensed under the Creative Commons Attribution 4.0 International License (CC BY 4.0), and we use the papers in accordance to their usage terms, with no private, sensitive, or personally identifiable information used in this work.

## LLM USAGE STATEMENT

In accordance with the ICLR 2026 policy on the use of Large Language Models (LLMs), we disclose that LLMs were used exclusively as general-purpose writing aids, such as for polishing grammar and improving readability. LLMs did not contribute to research ideation, experiment design, data analysis, or result interpretation, and thus played no role that would qualify as authorship or substantive contribution.

---

[2] https://info.arxiv.org/help/api/index.html
[3] https://docs.openreview.net/reference/api-v2
[4] https://aclanthology.org/

## A  AIRQA DATASET

### A.1  EXAMPLE FORMAT

Each data instance in AirQA is represented as a JSON dictionary containing the following fields:

- `uuid`: Globally unique uuid of the current task example.
- `question`: The user's question about the given papers.
- `answer_format`: The requirements for the output format of LLMs, e.g., "a Python list of text strings" or "a single float number", so that LLMs can form their answer accordingly and we can evaluate the answer conveniently.
- `tags`: A list of tags denoting different question types, element categories and evaluation types. Feasible tags include ["single", "multiple", "retrieval", "text", "image", "table", "formula", "metadata", "subjective", "objective"], corresponding to question types, modal categories in Section 2.1 and evaluation types in Section 2.2.
- `anchor_pdf`: A list of PDF uuids that are directly related to or explicitly mentioned in the question, provided in single and multiple questions.
- `reference_pdf`: A list of PDF uuids that may or may not help answer the question, only provided in multiple questions.
- `conference`: A list of conference name followed by year, provided in retrieval and comprehensive questions.
- `evaluator`: A dictionary containing 2 fields, `eval_func` and `eval_kwargs`, which defines how to evaluate the model outputs. Concretely,
  - the "`eval_func`" field defines the name of our customized Python function (or metric) which is used to compare the predicted result and the expected ground truth;
  - the "`eval_kwargs`" field defines the arguments for the corresponding evaluation function, which usually contain the gold or reference answer and other optional parameters.

### A.2  METADATA FORMAT

The metadata of each paper is organized in a structured JSON format, capturing key bibliographic and content-related attributes, as shown below:

- `uuid`: A universally unique identifier (UUID) assigned to this specific data sample.
- `title`: The full title of the academic paper.
- `conference_full`: The complete official name of the conference where the paper was published or presented. For example, "Annual Meeting of the Association for Computational Linguistics".
- `conference`: The standardized or abbreviated name of the conference, commonly used in citations or file naming. Examples include "ACL", "NeurIPS".
- `year`: The year in which the paper was published or presented at the conference.
- `volume`: The volume information of the conference proceedings or journal in which the paper appears, if applicable. For example, "NeurIPS 2023 poster".
- `bibtex`: A string containing the full BibTeX citation entry for the paper.
- `authors`: An ordered list of the authors who contributed to the paper.
- `pdf_url`: A direct URL linking to the downloadable PDF file of the paper. The link should point to an actual PDF file and therefore must end with ".pdf".
- `pdf_path`: The local file system path where the PDF is saved. The file should be renamed using the UUID to ensure consistent and collision-free naming.
- `num_pages`: An integer value indicating the total number of pages in the PDF document.
- `abstract`: The abstract of the paper, which is a concise summary of the research objectives, methodology, key findings, and implications.

- `tldr`: "Too Long; Didn't Read" summary — a brief, high-level summary of the paper's main contribution, typically one to two sentences.
- `tags`: A list of keywords or topic tags associated with the paper.

Table 8: The checklist of the 19 used evaluation functions, including their genres, categories, names, and descriptions.

| Genre | Category | Function | Description |
|---|---|---|---|
| objective | match | eval_bool_exact_match | Evaluate the output against the answer using exact boolean match. |
| | | eval_float_exact_match | Evaluate the output against the answer using exact float match with variable precision or tolerance. |
| | | eval_int_exact_match | Evaluate the output against the answer using exact integer match. |
| | | eval_string_exact_match | Evaluate the output against the answer using exact string match. |
| | | eval_string_fuzzy_match | Evaluate the output against the answer using fuzzy match provided by FuzzyWuzzy. |
| | | eval_structured_object_exact_match | Evaluate the output against the answer recursively by parsing them both as Python-style lists or dictionaries. |
| | set | eval_element_included | Evaluate whether the output is included in the answer list. |
| | | eval_element_list_included | Evaluate whether each element in the output list is included in the answer list. |
| | | eval_element_list_overlap | Evaluate whether the output list overlaps with the answer list. |
| | retrieval | eval_paper_relevance_with_reference_answer | Evaluate whether the retrieved paper is the same as the reference answer. |
| subjective | semantic | eval_reference_answer_with_llm | Evaluate the output against the reference answer using LLMs. |
| | | eval_candidate_reference_answer_with_llm | Evaluate whether the output matches any candidate reference answer. |
| | | eval_scoring_points_with_llm | Evaluate whether the scoring points are all mentioned in the output using LLMs. |
| | | eval_partial_scoring_points_with_llm | Evaluate whether the scoring points are partially mentioned in the output using LLMs. |
| | | eval_reference_answer_and_scoring_points_with_llm | Evaluate whether the reference answer and other scoring points are all mentioned in the output using LLMs. |
| | formula | eval_complex_math_formula_with_llm | Evaluate the mathematical equivalence between the output and the answer formatted in Latex using LLMs. |
| logical | | eval_conjunction | Evaluate the conjunction of multiple evaluation functions. The output passes the evaluation if and only if all the elements in the output pass the corresponding sub-evaluations. |
| | | eval_disjunction | Evaluate the disjunction of multiple evaluation functions. The output passes the evaluation if and only if at least one of the element in the output passes the corresponding sub-evaluation. |
| | | eval_negation | Evaluate the negation of an evaluation function. The output passes the evaluation if and only if it doesn't pass the original evaluation function. |

## A.3 Evaluation Function

In Table 8, we list the detailed names and descriptions of all 19 evaluation functions.

Here we present three representative cases corresponding to three distinct categories of evaluation functions: objective functions, subjective functions, and logical functions.

```
{
    "example": {
        "eval_func": "eval_string_exact_match",
        "eval_kwargs": {
            "gold": "Italian",
            "lowercase": true
        }
    }
}
```

Listing 1: Objective Function Case

In case (Listing 1), the evaluation function compares the predicted answer with the gold answer "Italian", ignoring case sensitivity due to the "lowercase" parameter being set to true. This means "italian", "ITALIAN", or "ItAliAn" would all match "Italian".

```
{
    "example": {
        "eval_func": "eval_reference_answer_with_llm",
        "eval_kwargs": {
            "reference_answer": "Artificial intelligence is a branch of
                ↪ computer science focused on building systems that can
                ↪ perform tasks that typically require human
                ↪ intelligence.",
            "question": "What is artificial intelligence?"
        }
    }
}
```

Listing 2: Subjective Function Case

In case (Listing 2), the evaluation checks if the predicted answer conveys the same meaning as the reference answer about artificial intelligence, in response to the question "What is artificial intelligence?".

The prompt used for conducting the subjective evaluation above is presented as an example among several possible formulations within this evaluation category, and is shown below.

> **Subjective Function Prompt Example**
>
> You are an intelligent judgement system who is expert in determining whether a predicted answer matches the reference answer in terms of semantic meaning and intent, based on the input question. You will be given the raw question, the reference answer, and the predicted answer. And you need to provide the final decision with the following format:
>
> ```txt
> True/False
> ```
>
> Notice that:
> 1. Remember to wrap the final judgement with triple backticks.
> 2. The final decision string must exactly be "True" or "False" without any extra character or punctuation. Any other text will be considered as incorrect.
> 3. The structure and format of the predicted answer do not matter. We only care about the semantic content, compared to the reference answer. Minor differences in grammar, structure, or formatting should be ignored if the core meaning is preserved.
> Now, let's start!
>
> [Question]: {question}
> [Reference Answer]: {reference_answer}
> [Predicted Answer]: {predicted_answer}
>
> Let's think step-by-step, and then provide the final judgement.

```
{
    "example": {
        "eval_func": "eval_disjunction",
        "eval_kwargs": {
            "eval_func_list": [
                "eval_string_exact_match",
                "eval_reference_answer_with_llm"
            ],
            "eval_kwargs_list": [
                {
                    "gold": "role-oriented routing",
                    "lowercase": true
                },
                {
                    "reference_answer": "It routes messages, requests,
                        ↪ or tasks based on the roles or
                        ↪ responsibilities of the recipients, rather
                        ↪ than simply by their identity or static
                        ↪ attributes.",
                    "question": "What's the most important idea of
                        ↪ role-oriented routing?"
                }
            ]
        }
    }
}
```

Listing 3: Logical Function Case

In case (Listing 3), the disjunction evaluation function checks if at least one of the specified evaluation functions returns a positive result. The first function, "eval_string_exact_match", verifies whether the predicted answer matches the gold standard "role-oriented routing" in a case-insensitive manner. The second function, "eval_reference_answer_with_llm", evaluates whether the predicted answer sufficiently addresses the question about the most important idea of role-oriented routing, as described in the provided reference answer. If either condition is satisfied, the evaluation returns 1; otherwise, it returns 0.

### A.4 ADDITIONAL DATASET STATISTICS

Here we include another two statistics on paper volume in Figure 5 and question lengths in Figure 6.

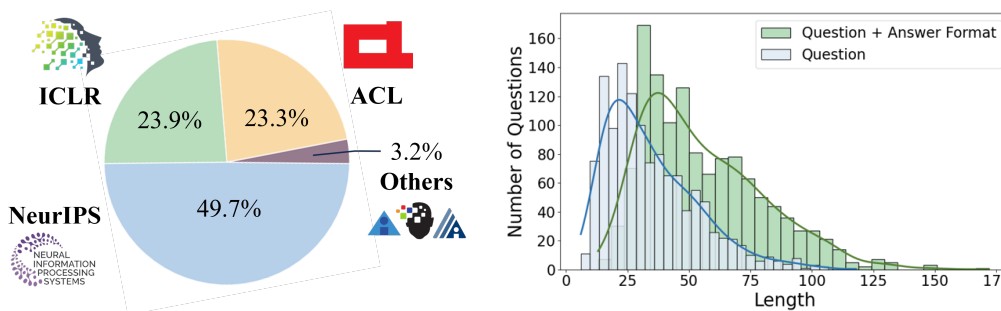

Figure 5: Conference distribution of the papers used.

Figure 6: Distribution of question and answer format lengths (in tokens) by count.

## B  AGENTIC BASELINE

In this section, we present detailed information of the actions implemented in the agentic baselines. Each action can be called in a Python-style manner, e.g., `Retrieve(query="Is there any work about the topic structured RAG?", limit=4)`

```
{
    "action_type": "Retrieve",
    "description": "Given a query text, retrieve relevant context from
        ↪ the Milvus vectorstore.",
    "observation": "The observation space is the retrieved top-ranked
        ↪ entries from the Milvus vectorstore based on queries.",
    "parameters": {
        "query": {
            "type": "str",
            "required": true,
            "description": "The query text will be encoded and used to
                ↪ search for relevant context. You can rephrase the
                ↪ original user question to obtain a more clear and
                ↪ structured query requirement."
        },
        "limit": {
            "type": "int",
            "required": false,
            "default": 5,
            "description": "The number of top-ranked context to
                ↪ retrieve. Please set it to a positive integer to limit
                ↪ the number of returned results. Extremely large limit
                ↪ values may be truncated."
        }
    },
    "use_cases": [
        {
            "example": {
                "query": "Is there any work about the topic structured
                    ↪ RAG?"
            },
            "explanation": "Retrieve top 5 pieces about a certain topic."
        },
        {
            "example": {
                "query": "What's the learning rate for training the
                    ↪ ResNet model?",
                "limit": 4
            },
            "explanation": "Retrieve detailed information about the
                ↪ learning rate for training the ResNet model. The top 4
                ↪ most relevant entries will be returned based on the
                ↪ query."
        }
    ]
},
{
    "action_type": "Query",
    "description": "Generate an SQL query to retrieve the desired
        ↪ information from the DuckDB database. Please refer to the
        ↪ concrete database schema to produce a valid and executable
        ↪ SQL.",
    "observation": "The observation space is the execution result of the
        ↪ SQL query. You do not need to worry about the actual
        ↪ execution, we will perform it for you. If the SQL failed to
        ↪ execute, we will return the error message. Extremely long SQL
        ↪ output will be truncated.",
    "parameters": {
        "sql": {
```

```
                "type": "str",
                "required": true,
                "description": "The concrete DuckDB SQL query to execute and
                    ↪ retrieve results."
            }
        },
        "use_cases": [
                {
                    "example": {
                        "sql": "SELECT abstract FROM metadata WHERE paper_id =
                            ↪ '12345678';"
                    },
                    "explanation": "Get the abstract of the paper with paper_id
                        ↪ '12345678' from the metadata table in the DuckDB
                        ↪ database."
                },
                {
                    "example": {
                        "sql": "SELECT pages.page_number FROM images JOIN pages
                            ↪ JOIN metadata ON images.ref_page_id =
                            ↪ pages.page_id AND pages.ref_paper_id =
                            ↪ metadata.paper_id WHERE metadata.paper_id =
                            ↪ '12345678' AND images.image_caption LIKE '%Figure
                            ↪ 3%';"
                    },
                    "explanation": "Find which page in the paper with paper_id
                        ↪ '12345678' contains Figure 3."
                }
        ]
    },
    {
        "action_type": "Answer",
        "description": "When you take this action, the retrieved results
            ↪ suffice to answer the user question. PLEASE STRICTLY ADHERE TO
            ↪ THE ANSWER FORMAT FOR THE CURRENT QUESTION.",
        "observation": "There is no observation for this terminal action,
            ↪ since it indicates the completion of the task and end of the
            ↪ interaction.",
        "parameters": {
            "answer": {
                "type": "Any",
                "required": true,
                "description": "The final answer to the user question."
            }
        },
        "use_case": [
                {
                    "example": {
                        "answer": 42
                    },
                    "explanation": "The final answer is 42."
                },
                {
                    "example": {
                        "answer": ["Results", "Discussion"]
                    },
                    "explanation": "The final answer is a list of strings:
                        ↪ ['Results', 'Discussion']."
                }
        ]

}
```

Listing 4: Detailed Action Format

# C   SUPPLEMENTARY EXPERIMENTS AND SETTINGS

## C.1   DETAILED HYPER-PARAMETERS FOR INSTRUCTION TUNING

Table 9: Hyper-parameters for Instruction Tuning.

| Hyper-Parameter | Default Value |
| --- | --- |
| Finetuning Type | LoRA |
| LoRA Target | all |
| LoRA Rank | 16 |
| LoRA Alpha | 16 |
| LoRA Dropout | 0.05 |
| Cutoff Length | 4,096 |
| Mask History | true |
| Gradient Accumulation Steps | 16 |
| Learning Rate | $1 \times 10^{-4}$ |
| Train Epochs | 1.0 |
| Learning Rate Scheduler | Cosine |
| Warmup Ratio | 0.1 |

## C.2   PREPROCESSING

Besides collecting papers and metadata illustrated in Section 2.3, we parse the papers with PyMuPDF (Artifex Software, 2023) and `MinerU` (Wang et al., 2024), and populate relevant information into the relational database DuckDB (Mühleisen & Raasveldt, 2024). Additionally, we segment raw documents into chunks of 512 tokens, encode the chunks with `all-MiniLM-L6-v2` (Wang et al., 2020), and insert the vectors into the vectorstore Milvus (Wang et al., 2021).

## C.3   ABLATION ON VLMS

To further investigate whether direct access to visual information improves performance, we implement an additional action, "View", for the Agentic Hybrid baseline.

```
{
    "action_type": "View",
    "description": "You can retrieve the visual information of the paper
        ↪ by taking this action. Please provide the paper id, the page
        ↪ number, and the optional bounding box.",
    "observation": "The observation space is the image that you want to
        ↪ view. We will show you the image according to your parameters.
        ↪ The error message will be shown if there is any problem with
        ↪ the image retrieval.",
    "parameters": {
        "paper_id": {
            "type": "str",
            "required": true,
            "description": "The paper id to retrieve the image."
        },
        "page_number": {
            "type": "int",
            "required": true,
            "description": "The page number (starting from 1) of the
                ↪ paper to retrieve the image."
        },
        "bounding_box": {
            "type": "List[float]",
            "required": false,
```

```
        "default": [],
        "description": "The bounding box of the image to retrieve.
            ↪ The format is [x_min, y_min, delta_x, delta_y]. The
            ↪ complete page will be retrieved if not provided."
    }
},
"use_cases": [
    {
        "example": {
            "paper_id": "12345678",
            "page_number": 3,
            "bounding_box": []
        },
        "explanation": "Retrieve the image of the third page of the
            ↪ paper with id 12345678."
    },
    {
        "example": {
            "paper_id": "12345678",
            "page_number": 5,
            "bounding_box": [
                51.1,
                204.3,
                333.0,
                13.8
            ]
        },
        "explanation": "Retrieve the image of the fifth page of the
            ↪ paper with id 12345678, with a bounding box of [51.1,
            ↪ 204.3, 384.1, 218.1]."
    }
]
}
```

Listing 5: Details for "View" action.

Through this action, LLMs can **access the image content encoded in base64 format** by specifying the paper ID, page number, and the bounding box of the relevant region.

Table 10: Performance of Agentic Hybrid with "View" action on AirQA dataset.

| Size | "View"? | text | table | image | form. | meta. | AVG |
|------|---------|------|-------|-------|-------|-------|-----|
| 32B | ✗ | 31.88 | 10.33 | 17.87 | 18.90 | 21.31 | 24.24 |
|     | ✓ | 31.56 | 8.92 | 20.77 | 15.75 | 25.41 | 24.56 |
| 72B | ✗ | 36.55 | 15.96 | 23.67 | 20.47 | 36.88 | 30.34 |
|     | ✓ | 36.55 | 15.96 | 24.64 | 22.83 | 36.07 | 30.78 |

We conduct experiments with Qwen2.5-VL family (Bai et al., 2025) as backbone model. Table 10 shows that, with direct access to images, the performance on visual questions improves, though further methods for enhancement remain to be explored.

## C.4 FINE-TUNING BASED ON AIRQA

To examine the relative effectiveness of synthetic data compared to manually annotated data, we fine-tune another model with examples directly from the AirQA dataset. Due to the unquantifiable nature of manually annotated trajectories, we continue to use Qwen2.5-32B-Instruct as teacher model for trajectory generation.

Table 11 indicates that, model fine-tuned on 400 examples from AirQA achieves performance comparable to that trained on 1,000 automated generated examples. It is worth noticing that while

Table 11: Comparison between models fine-tuned on examples directly from the AirQA dataset and examples generated by the EXTRACTOR.

| Dataset | Count | sgl. | multi. | retr. | comp. | AVG |
|---|---|---|---|---|---|---|
| - | - | 16.24 | 3.72 | 26.39 | 15.85 | 15.24 |
| AirQA | 400 | 15.67 | 4.95 | 48.26 | 13.73 | 19.98 |
| EXTRACTOR | 1000 | 15.95 | 2.48 | 48.96 | 14.44 | 19.74 |

manual examples seem to be more effective, they come at a higher cost. It takes 20 minutes for a human to generate an example and only 20 seconds for EXTRACTOR.

## C.5 HUMAN STUDY

To provide a reference for the difficulty of the AirQA dataset, we recruit 3 students with expertise in artificial intelligence to answer 98 questions sampled from our AirQA dataset. They are strictly prohibited from using any form of LLM and are only allowed to search the internet. Each question has a time limit of 20 minutes.

Table 12: Performance of human experts on AirQA dataset.

| Question Type | | | | Element Category | | | | | Evaluation | | AVG |
|---|---|---|---|---|---|---|---|---|---|---|---|
| sgl. | multi. | retr. | comp. | text | table | image | form. | meta. | obj. | subj. | |
| 64.29 | 54.00 | 52.17 | 56.82 | 53.12 | 56.07 | 67.05 | 50.00 | 55.00 | 58.52 | 52.58 | 56.63 |

Results in table 12 show that AirQA is a highly challenging dataset, even human experts are only able to achieve a relatively high score within the time limit, rather than a perfect one. Meanwhile, all three participants report difficulty in identifying the correct papers, particularly when multiple sources are involved, and two additionally note challenges in understanding domain-specific terminology.

## C.6 STATISTCS ON TIME & COST

**Evaluation** Here we provide an empirical estimate: evaluating all 1,246 examples takes approximately 35 minutes and costs $0.056. This process can be further accelerated using simple parallelization techniques if needed.

**Agentic Baselines** In Table 13 we list the time and cost per example for two models and three agentic baselines for reference.

Table 13: Statistics of the number of interaction(s), accumulated prompt / completion token(s), time consumption, and LLM cost per sample with different models and agentic methods on AirQA.

| Model | RAG Method | # Turn(s) | # Prompt Token(s) | # Completion Token(s) | Time (s) | Cost ($) |
|---|---|---|---|---|---|---|
| Qwen2.5-72B-instruct | Agentic RAG | 7.53 | 39870 | 658 | 58 | - |
| | Agentic Text2SQL | 6.45 | 42991 | 790 | 70 | - |
| | Agentic Hybrid | 5.95 | 62533 | 767 | 62 | - |
| GPT-4o | Agentic RAG | 4.59 | 13231 | 365 | 13 | 0.0367 |
| | Agentic Text2SQL | 7.26 | 32957 | 815 | 22 | 0.0905 |
| | Agentic Hybrid | 5.08 | 35909 | 566 | 18 | 0.0954 |

**Example Synthesis** Though we have discussed in App. C.4 that LLM-based methods are far more efficient than manual annotation, we still believe that a rough cost analysis would be helpful to offer an empirical reference for future work.

The results for each question type are computed by averaging the time and cost of generating 10 examples with GPT-4.1-mini. A back-of-the-envelope calculation indicates that it costs roughly

Table 14: Average time and cost for synthesizing an example. The time and cost for "multiple" question type is estimated by directly doubling that of "single".

|        | Time (s) | Cost ($) |
|--------|----------|----------|
| sgl.   | 18.6     | 0.041    |
| multi. | 37.2     | 0.082    |
| retr.  | 5.2      | 0.002    |
| comp.  | 18.5     | 0.039    |

$160 and 22 hours to produce $4,000$ examples ($1,000$ per question type). Given the complexity of the tasks, this is reasonably efficient compared with human annotation, and can be further improved by using open-source models or parallelization.

## D    SUPPLEMENTARY ANALYSIS

### D.1    ERROR ANALYSIS ON GPT-4O

To further illustrate the bottleneck of our AirQA dataset, we randomly sample 60 examples (15 for each question type) where GPT-4o + Agentic Hybrid produces incorrect answers. Through manual analysis, we identify the following five root causes that ultimately lead to mistakes:

1. **Lack of Context**: The agent fails to use the given tools to find relevant snippets.
2. **Over Confidence**: The agent chooses to generate the answer too early.
3. **Missing Paper**: For questions that involve multiple papers, the agent fails to realize/find other papers.
4. **Textual Reasoning**: The agent successfully retrieves the key snippet but fails to understand it.
5. **Visual Reasoning**: The agent fails to retrieve/understand paratextual information.

Table 15: Error Analysis of Agentic Hybrid baseline with GPT-4o as backbone.

| Category | sgl. | multi. | retr. | comp. | Total |
|---|---|---|---|---|---|
| Lack of Context | **5** | 3 | **9** | **6** | **23** |
| Over Confidence | 3 | 2 | 5 | 3 | 13 |
| Textual Reasoning | 2 | 1 | 1 | 5 | 9 |
| Missing Paper | 0 | **8** | 0 | 0 | 8 |
| Visual Reasoning | **5** | 1 | 0 | 1 | 7 |

The analysis shows that the dominant cause differs across question types, while generally, current models still exhibit limitations in **long-term planning and multi-modal reasoning**, suggesting the need for new methods that better balance planning and acting.

### D.2    ERROR ANALYSIS ON FINE-TUNED 7B

Similarly, we conduct an error analysis on fine-tuned Qwen2.5-7B-Instruct. Besides the aforementioned five causes, we identify another reason that ultimately leads to failures:

6. **Repetition**: The agent consistently predicts the same action.

Table 16: Error Analysis of Agentic Hybrid baseline with fine-tuned 7B as backbone.

| Category | sgl. | multi. | retr. | comp. | Total |
|---|---|---|---|---|---|
| Lack of Context | 4 | **4** | **9** | 3 | **20** |
| Textual Reasoning | **6** | 2 | 3 | 2 | 13 |
| Over Confidence | 2 | 3 | 2 | 3 | 10 |
| Repetition | 1 | 3 | 1 | **4** | 9 |
| Visual Reasoning | 2 | 1 | 0 | 3 | 6 |
| Missing Paper | 0 | 2 | 0 | 0 | 2 |

The analysis indicates that fine-tuned models still exhibit limitations in **textual retrieval and comprehension**, highlighting areas for future improvement.

### D.3    IMPROVEMENT ANALYSIS ON 7B

We further investigate which aspects of small models are improved by EXTRACTOR. We sample 10 examples for each question type where fine-tuned Qwen2.5-7B-Instruct outperform untrained baseline, totaling 40 examples. For each example, we examine the source of improvement and categorize it into one of three main contributing factors:

1. **Better Retrieval Strategy**: The model develops a clearer understanding of the overall retrieval process and the specific actions required at each step, allowing it to **plan more effectively** and identify useful intermediate steps.

2. **Better Retrieval Behavior**: The model interacts more accurately with the environment, correctly interpreting schemas and providing appropriate tool parameters.

3. **Better Understanding and Reasoning**: The model demonstrates deeper comprehension of the question and context, producing more coherent and reliable reasoning.

Table 17: Improvement Analysis of Fine-Tuned 7B.

| Category | sgl. | multi. | retr. | comp. | Total |
|---|---|---|---|---|---|
| Strategy | **4** | **7** | 1 | 1 | 13 |
| Behavior | 3 | 1 | 4 | **6** | **14** |
| Understanding | 3 | 2 | **5** | 3 | 13 |

The analysis shows that the dominant contributor varies across question types. While improved retrieval behavior is not always the primary factor, **reductions in tool usage errors are consistently observed**, enhancing both retrieval behavior and strategy as the model learns to use tools more effectively. This finding aligns with the component ablation in Section 4.3 and reinforces the claim that **our training framework enhances the model's ability to generate valid actions**.

# E  SYNTHESIS PROMPT

## E.1  EXPLORER PROMPT

Here we showcase prompt templates of explorers discussed in Section 3.2.

---

**Explorer Prompt for Single and Comprehensive question types**

You are an intelligent annotation system who is expert in posing questions.
{description}
Your output should be in the following format:
[Thought]: Your thought process.
```txt
[Question]: Your question here.
[Answer]: Your answer here.
```

Notice that:
- Remember to wrap your output (except [Thought]) with triple backticks.
- Don't include the answer in the question or in the reasoning steps.
- Your question should be as objective as possible.
- Your answer should be concise and clear.
{hint}
Let's think step-by-step, and then provide the final question and answer.
{context}

---

Here, {description} stands for the description of the task, {hint} includes additional hints, and {context} represents the corresponding context for question generation.

For example, for single question type and table element category, the description is *"You will be given an AI research paper, and your task is to generate a question based on the content of the table in HTML format and the caption of the table."*, the hint prompt is *"- Try not to include the word 'table' in your question."*, and the context prompt is:

---

**Context Prompt for Single type, Table category**

The caption of the table is as follows:
```txt
{caption}
```

The content of the table is as follows:
```html
{content}
```

---

where the caption and the content are the raw text caption and the table content in HTML format respectively.

---

**Explorer Prompt for Retrival question type**

You are an intelligent annotation system who is expert in posing questions. You need to pose a question based on the title and abstract of a paper, where the answer to the question should be the title of the paper. That is to say, you need to describte the contribution or the feature of the paper in the question, so that the respondents can identify the paper. Don't include the title itself in the question. Now let's start!

[Title]: {title}
[Abstract]: {abstract}

Your output should be in the following format:

Your thought process.
```txt
Your question here.
```

---

> Note that, you should wrap your output with triple backticks.

For retrieval question type, we use the different explorer prompt shown above, and for multiple question type, there is no need of an explorer, as we simply combine two single questions.

### E.2 TRACKER PROMPT

Here we present prompt templates of trackers discussed in Section 3.3.

---

**Tracker Prompt for Single and Comprehensive question types**

You are an intelligent annotation system who is expert in reviewing questions.

You will be given a question and an answer. You should adjust the question and the answer, adapting them to the evaluator's requirements. The descriptions, parameters and use cases of the evaluators are provided below:

________________________________________________

{evaluator}

Note that:
- If you want the predicted answer list to be exactly same with the gold answer list, use `eval_structured_object_exact_match`, don't use `eval_element_list_included`.
- If your evaluation involves list matching, and the order doesn't matter, set `ignore_order` to `true`. If the order matters, set `ignore_order` to `false`.
- If you are sure that the answer is unique, there aren't other equivalent answers, and any rephrase will change the semantic meaning of the answer, you can use `eval_string_exact_match`. Otherwise, you should use `eval_reference_answer_with_llm`. Generally, we recommend using `eval_reference_answer_with_llm` for subjective questions, and `eval_string_exact_match` for single-word answers.

________________________________________________

Your output should be in the following format:
[thought]: Your thought process.
```txt
[question]: Modified question.
[evaluator]: The evaluator you choose.
[answer_format]: The format that the respondent should follow in order to pass the evaluator. e.g. "Your answer should be a single python list containing two strings, the first element of the list is the abbreviation of the baseline, the second element of the list is the full name of this baseline, e.g.["abbr","full"].".
[answer]: Modified answer.
[tag]: A single `subjective` or `objective` without explanation. Whether the evaluator involves LLM. `subjective` if it involves LLM, otherwise `objective`.
```

Note that:
- Remember to wrap your output (except thought) with triple backticks.
- DON'T INCLUDE ANSWERS, HINTS OR KEY POINTS IN [question] OR [answer_format] IN ANY FORM, ESPECIALLY WHEN YOU TRY TO ILLUSTRATE [answer_format] BY GIVING EXAMPLES.
- [answer_format] will be provided to the respondent along with the [question]. [question] and [answer_format] together form the who question that will be presented to the respondent. [question] focuses on the question itself, [answer_format] focuses on the format of the answer.
- You should present [evaluator] in JSON format, as given in the use cases. And your [answer] should be able to pass the evaluator.
- You can modify the question and answer based on the evaluator's requirements, but don't change the original meaning of the question and answer.
- When the question involves percentage, and the percentage is an exact value, not an approximate value, try to use `eval_float_exact_match` or `eval_int_exact_match`, while indicating the decimal places in [answer_format].

Here're the original question and answer:
```txt

---

> [question]: {question}
> [answer]: {answer}
> ```
>
> Let's think step-by-step, and then provide the final arguments.

Here, {evaluator} contains the detailed information of the 19 evaluation functions, and {question} and {answer} stand for the question-answer pair generated by the explorer. The evaluator prompt is in the following format:

> **Tracker Prompt for Single and Comprehensive question types**
>
> ## {function}
>
> ### Description
> {description}
>
> ### Parameters
> {parameters}
>
> ### Use Case(s)
> {use_case}

which contains the name, the description, the parameters and the use cases of the functions.

## F    LIMITATIONS AND BROADER IMPACTS

Although precise question answering datasets AirQA for academic papers can enhance research efficiency, this work still has certain limitations: 1) As large language models incorporate more academic papers during pre-training, some questions can be answered solely based on their parametric knowledge; 2) The current dataset is limited to English-language papers in the field of artificial intelligence, and its coverage remains to be improved; 3) While most questions can be evaluated using objective scoring functions, long-form answers inevitably rely on large model-based evaluation, which may affect the consistency and stability of the evaluation results as the continual training and update of these LLMs. For broader social impact, as LLM-based agents become increasingly robust and practical through more refined agent-level finetuning, their improved question-answering capabilities can help researchers save significant time on literature review and detail retrieval, avoid reinventing the wheel, and even assist in building personalized knowledge bases of academic papers.

