# OpenReview forum: "AirQA: A Comprehensive QA Dataset for AI Research with Instance-Level Evaluation"
_ICLR.cc/2026/Conference — ICLR 2026 Poster_

### Official Review · Reviewer_ukJ4 · 2025-10-21

**Soundness:** 4
**Presentation:** 4
**Contribution:** 2
**Rating:** 4
**Confidence:** 3

**Summary:**

The authors introduce M4PQA, a human-annotated comprehensive paper QA dataset in the field of artificial intelligence, with 13,948 papers and 1,246 questions, that encompasses multi-task, multi-modal and instance-level evaluation. They also introduce EXTRACTOR, an automated framework for instruction data synthesis which can perform example generation and trajectory collection without human intervention.

**Strengths:**

- Expert annotated from students with AI expertise (n=26).

- Expert validated study for difficulty (n=3).

- Paper well written.

- Experiments well designed and good ablations.

- Authors address shortcomings of other scientific papers benchmarks.

- Authors show that performance raises consistently as data scales up, highlighting the scalability of the framework.

- Function-based evaluation into QA domain which is systematic and reproducible.

**Weaknesses:**

- Even though the work is good and addresses gaps in other datasets I am not confident how significant is the novelty compared to other similar works. Based on table 3, one can run on different benchmarks from the literature to assess on different question types (Single, Multi, Retrieval, ..., Image Form. Meta). This is the main reason for my score.

- How is extractor validated? Any human intervention?

- Maybe if the annotators had more specific instructions like identifying the key components of the papers and coming up with key questions would improve the quality of questions in terms of usefulness? Perhaps identifying questions that require some kind of thinking/logical deduction?

**Questions:**

- Any reason not including ICML, AAAI or big vision conferences (e.g., CVPR)?

- Small typo: figure 3 “based on it”

- The main purpose of extractor is to automatically produce questions/trajectories/answers that are then used to finetune the LLM on these tasks? If you want to use it to extend your proposed dataset you still need a human expert in the loop to make sure the questions/trajectories/answers are correct.

- Did you perform SFT only? Any RL methods that can be potentially be used (e.g., GRPO)?

- Did you use the same papers as your dataset with extractor to generate questions/trajectories/answers? Which model was used for generating these trajectories?

---

> ### Author Response · Authors · 2025-11-25
> **Rebuttal to Reviewer ukJ4 (1/2)**
>
> We would like to sincerely thank the reviewer for your kindness, valuable advice, and precious time in reviewing our work.
>
> **Q1: Contribution of M4PQA**
>
> **A1**: We appreciate the reviewer for allowing us to highlight the contributions of our dataset. We would like to illustrate its novelty from the following two perspectives:
>
> 1. **Combination leads to Difficulty**
>
> - In many domains, a complex task can be decomposed into a sequence of atomic tasks, and it is a common practice to tackle these simpler tasks first. However, solving the atomic tasks does not guarantee that the complex task is also solved. This is why “Needle-in-a-Stack” remains a challenging problem even when LLMs handle short contexts well, and why long-term reasoning is still underexplored.
>
>     In question answering, a similar observation holds: single-hop or single-document QA is largely solved, whereas multi-hop and multi-document QA remain underexplored, with limited benchmark resources and datasets.
>
> - We argue that a well-designed scientific QA system should be capable of:
>
>     - Answering questions based on both textual and paratextual elements of a paper.
>     - Retrieving relevant papers given a specific description.
>     - Applying long-term planning and reasoning to orchestrate the execution of the previous two capabilities.
>
>     The first two correspond to atomic tasks (single and retrieval question types), while the third capability is crucial for solving complex tasks, as it enables iterative and alternating execution of the atomic tasks required by the “multiple” and “comprehensive” question types in our dataset.
>
> - While we mark some datasets as containing “multiple” and “comprehensive” questions in Table 3, they or their simple combinations fall short in fully covering the three capabilities outlined above. Concretely:
>
>     - M3SciQA adopts a rule-based example construction with fixed question templates, offering little room for flexible reasoning and planning.
>     - SciDQA provides references directly for “multiple” question types, bypassing the need for paper retrieval, which is a core component of our “multiple” questions (line 147).
>     - LitQA2 emphasizes text-based retrieval but largely overlooks paratextual elements.
>
>      This persistent neglect of task combination demonstrates the novelty of M4PQA: it requires models to perform **long-term, multi-modal reasoning and planning** under realistic scientific QA scenarios.
>
> 2. **Fact-Oriented Evaluation**
>
> - We emphasize that the evaluation framework introduced in Section 2.2 represents an important novelty. While most previous QA datasets rely on linguistic metrics or LLM-based assessment, which prioritize semantic coherence over factual correctness, our framework primarily focuses on evaluating the **factual accuracy** of the answers.
>
> - To this end, we employ **instance-level evaluation**, assessing each question individually to enable fine-grained, example-specific scoring. In addition, we use **output reformatting**, providing a structured output format for each question (e.g., a Python list of integers) to guide models in producing the elements that are directly evaluated.
>
> - To the best of our knowledge, both instance-level evaluation and output reformatting are the first of their kind applied in the QA domain.

---

> > ### Author Response · Authors · 2025-11-25
> > **Rebuttal to Reviewer ukJ4 (2/2)**
> >
> > **Q2: Implementation Details of ExTrActor**
> >
> > **A2**: We notice that the reviewer has several questions regarding our ExTrActor framework, we would like to answer them one by one according to their importance.
> >
> > 1. **Quality Analysis of Generated Questions**
> >
> > - We do not evaluate any method on ExTrActor-generated examples; they are used only to produce trajectories for training. All evaluations are conducted on M4PQA to provide a straightforward comparison between fine-tuned models and baseline models with different methods.
> >
> > - Regarding the reliability of ExTrActor-generated examples, results on M4PQA (Table 6) already show that these synthetic examples help improve performance without human annotations, suggesting their adequacy for fine-tuning. But we agree that further analysis of LLM-generated examples could strengthen this claim.
> >
> > - We sampled 120 synthetic examples and evaluated them with DeepSeek-V3.1 along three aspects: **format**, **correctness**, and **difficulty**, obtaining average scores of 4.5, 3.2, and 2.6 out of 5, respectively. A manual check on a subset further shows that its judgments are largely consistent with human assessments, indicating that although these examples are not very challenging, they remain reliable for fine-tuning.
> >
> > - In practice, any human intervention would substantially limit the scalability of this method. As noted, manual verification is necessary to ensure that examples are valid for inclusion in the test set. For this reason, we rely exclusively on human-annotated examples in M4PQA.
> >
> > 2. **RL-based Methods**
> >
> > We understand and acknowledge your opinion that LRM represents a current research trend, especially in multi-turn environments. A member of our team is currently exploring this area by **enhancing models’ multi-step reasoning capabilities through reinforcement learning**. We aim to refine this work further, so detailed and complete experimental results are not included in this paper. We provide below a subset of preliminary results obtained using GRPO:
> >
> > |Method|sgl.|multi.|retr.|comp.|AVG|
> > |---|---|---|---|---|---|
> > |SFT|21.08|4.95|51.04|22.18|24.07|
> > |SFT + GRPO|22.22|6.19|52.78|20.42|24.24|
> >
> > Training is conducted on Qwen2.5-7B-Instruct, and results demonstrate that reinforcement learning yields further performance improvement within just 300 training steps. We will continue to advance our work in this direction in future research.
> >
> > 3. **Additional Details**
> >
> > - **Paper Usage**: When choosing papers for example synthesis, we make every effort to avoid overlap with M4PQA. However, since manual annotation and automatic synthesis proceed in parallel, complete certainty regarding overlap cannot be guaranteed.
> >
> > - **Model Usage**: As noted in Section 4.2 (line 449), all three agents in ExTrActor use Qwen2.5-32B-Instruct as backbone model.
> >
> > - **Explorer Implementation**: Detailed information is critical for daily research. While previous methods and datasets often focus only on key points, we consider comprehensive  details to be valuable. Therefore, we do not explicitly distinguish between key points and other details. Regarding questions requiring reasoning or logical deduction, our design includes a setting where two relevant sections are provided together, and Explorer is tasked with generating two pertinent questions based on these sections.
> >
> > **Q3: Volume Selection for M4PQA**
> >
> > **A3**: The retrieval scale mentioned in line 215 (ACL2023, ICLR2024, and NeurIPS2024) applies only to anchor papers for the retrieval and comprehensive question types. For the other question types, all papers are accepted, although annotators are encouraged to prioritize papers from these three conferences to facilitate preprocessing. As shown in Figure 5, papers from other conferences are also included, but their proportion is relatively small since we count all papers from ACL2023, ICLR2024, and NeurIPS2024 as used.
> >
> > **Q4: Typo**
> >
> > **A4**: We would like to sincerely thank the reviewer for your carefulness. We have fixed these typos.

---

> > > ### Comment · Reviewer_ukJ4 · 2025-11-26
> > >
> > > Thanks for clarifying the task combination and the instance-level evaluation which are a differentiating factor from other related works. I do agree that this is an important evaluation component that previous works do not have. This was my main concern and I will be raising my score.
> > >
> > > I would also suggest to quantify the overlap between the synthesized examples and M4PQA in your ExTrActor framework. It would be good to make sure that the performance gain is not due to any data contamination. Ideally training and testing set should come from different papers to minimize this risk.

---

> > > > ### Author Response · Authors · 2025-11-29
> > > >
> > > > Thanks very much for your acknowledgement. Our overlap analysis shows that only 15 ExTrActor-generated questions reach a cosine similarity above 0.8 with any question in M4PQA, and only about 1% of the papers used in M4PQA appear in the synthetic examples produced by ExTrActor, suggesting minimal risk of data contamination.

---

### Official Review · Reviewer_b9vj · 2025-10-31

**Soundness:** 3
**Presentation:** 3
**Contribution:** 3
**Rating:** 6
**Confidence:** 4

**Summary:**

This paper introduces M4PQA, a question-answering dataset containing 1,246 questions derived from documents in the field of artificial intelligence. The dataset is designed to evaluate the capabilities of large language model (LLM) agents in answering questions about research papers. It was manually curated by students with AI expertise and includes four categories of questions such as single-document, multiple-document, retrieval, and comprehensive, and targets different components of research papers (e.g., text, tables, figures). The authors demonstrate that state-of-the-art models such as GPT-4o and Qwen2.5 struggle to effectively answer questions from M4PQA under various prompting and agentic settings. Furthermore, the paper proposes Extractor, a framework that leverages LLMs to generate synthetic QA trajectories for agentic fine-tuning.

**Strengths:**

- The paper addresses a well-motivated and relevant problem.
- The proposed dataset is comprehensive, encompassing a diverse range of question types and context variations.
- The evaluation is thorough, testing the dataset across multiple models and agentic approaches.

**Weaknesses:**

- The effectiveness of the proposed extractor framework is not convincingly demonstrated. Since the framework is entirely LLM-based, its reliability and the quality of the generated questions are uncertain. The paper lacks dedicated experiments evaluating the quality of the synthetic trajectories produced. In addition, Table 11 indicates potential quality issues with the Extractor framework. A comparison with other automatic question-generation methods could have provided stronger empirical support.
- If scalability is a primary motivation for introducing the extractor framework, a cost or efficiency analysis should be included to support this claim.

**Questions:**

- How reliable are the LLM-based (subjective) evaluation functions? Additionally, since LLM-based evaluations can be computationally expensive, doesn’t this approach introduce a significant cost whenever a method is evaluated on M4PQG? It would be helpful if you could provide an approximate cost estimate.
- From line 430, it appears that Table 6 presents the results of the extractor framework. If this is correct, could you clarify why the caption currently states M4PQG?
- In Section 3.2, could you elaborate on how the context extraction process is performed?

---

> ### Author Response · Authors · 2025-11-25
> **Rebuttal to Reviewer b9vj (1/2)**
>
> We would like to sincerely thank the reviewer for your kindness, valuable advice, and precious time in reviewing our work.
>
> **Q1: Implementation Details of ExTrActor**
>
> **A1**: We acknowledge that the reviewer raised several questions regarding the ExTrActor framework, and we address them in order of importance.
>
> 1. **Evaluation of ExTrActor**
>
> We do not evaluate any method on ExTrActor-generated examples; they are used only to produce trajectories for training. All evaluations are conducted on M4PQA to provide a straightforward comparison between fine-tuned models and baseline models with different methods. We have revised the caption of Table 6 to make sure readers understand that the results are by models trained using ExTrActor and evaluated on M4PQA.
>
> 2. **Quality Analysis of Generated Questions and Trajectories**
>
> - Evaluation results on M4PQA (Table 6) already show that the generated examples and trajectories improve performance without human intervention, indicating that they are sufficient for fine-tuning. Still, we agree with the reviewer that a dedicated analysis of LLM-generated questions and trajectories could further strengthen this conclusion.
>
> - We sampled 120 synthetic examples and evaluated them with DeepSeek-V3.1 along three aspects: **format**, **correctness**, and **difficulty**, obtaining average scores of 4.5, 3.2, and 2.6 out of 5, respectively. A manual check on a subset further shows that its judgments are largely consistent with human assessments, indicating that although these examples are not very challenging, they remain reliable for fine-tuning.
>
> - A similar analysis was performed on 120 sampled trajectories, evaluated on **tool usage** and **logical reasoning**, which achieved average scores of 3.6 and 3.4 out of 5, respectively. These results indicate that the trajectories exhibit reasonably good structure and can offer useful patterns for smaller models to learn from.
>
> - We would also like to clarify that although Table 11 shows that M4PQA examples are more efficient for training, it is important to note that **a more challenging question is often more beneficial for fine-tuning than a simpler one**. Thus, the relatively lower efficiency does not imply reduced usefulness, it simply reflects that the question is easier. In fact, as discussed in Appendix C.5 (line 1177) and acknowledged by the reviewer, ExTrActor remains a more scalable alternative for instruction data generation.
>
> 3. **Other Question-Generation Methods**
>
> As discussed in sub-question 2, we use generated questions solely for sampling agentic trajectories for training. Experimental results also show that medium-quality questions can still effectively and efficiently improve the performance of smaller models.
>
> To the best of our knowledge, SciQAG is the only document-based question-generation method for scientific papers. However, its methodology is largely similar to Explorer. The resulting QA pairs are used directly for instruction tuning without further processing, whereas our framework employs **an Actor to generate interaction trajectories** for instruction tuning, which provides a fundamentally different training signal.
>
> Similar to SciQAG, we also experiment with directly using QA pairs w/o CoT as training data:
> |Method|sgl.|multi.|retr.|comp.|AVG|
> |---|---|---|---|---|---|
> |/|7.12|1.55|0.00|7.04|4.01|
> |QA|8.26|1.24|0.00|5.99|4.01|
> |- w/ CoT|7.69|1.24|0.00|5.28|3.69|
>
> Results above show that **fine-tuning on QA pairs does not lead to any observable improvement on our M4PQA dataset**.
>
> 4. **Cost and Efficiency Analysis**
>
> We understand the reviewer’s point that although LLM-based methods are far more efficient than manual annotation, a rough cost analysis would still be helpful to offer an empirical reference for future work.
>
> |          | time (s) | cost ($) |
> | -------- | -------- | -------- |
> | sgl.     | 18.6     | 0.041    |
> | (\*)multi. | 37.2     | 0.082    |
> | retr.    | 5.2      | 0.002    |
> | comp.    | 18.5     | 0.039    |
>
> \* the time and cost for “multiple” question type is estimated by doubling that of “single”
>
> The results for each question type are computed by averaging the time and cost of generating 10 examples with GPT-4.1-mini. A back-of-the-envelope calculation indicates that it costs roughly $160 and 22 hours to produce 4,000 examples (1,000 per question type). Given the complexity of the tasks, this is reasonably efficient compared with human annotation, and can be further improved by using open-source models or parallelization.

---

> > ### Author Response · Authors · 2025-11-25
> > **Rebuttal to Reviewer b9vj (2/2)**
> >
> > 5. **Context Extraction for Explorer**
> >
> > We use MinerU for extracting paratextual elements. Given a PDF, MinerU automatically identifies structural components such as tables and figures. For example, a parsed table is returned in the following format:
> >
> > ```json
> > {
> >     "type": "table",
> >     "img_path": "images/xxxx.jpg",
> >     "table_caption": ["the caption of the table"],
> >     "table_footnote": ["the footnote of the table"],
> >     "table_body": "the table in HTML format",
> >     "page_idx": 6
> > },
> > ```
> >
> > We then incorporate this information as context into carefully designed prompts (see Appendix E.1) to guide Explorer to generate natural language QA pairs based on the extracted content.
> >
> > **Q2: Implementation Details of Instance-Level Evaluation**
> >
> > **A2**: We appreciate the reviewer’s attention to our subjective evaluation functions. We would like to provide further illustrations from the perspective of reliability and efficiency:
> >
> > 1. **Reliability**
> >
> > As we mention in Appendix A.3,
> >
> > - While LLM-based judgment is necessary in certain cases, we design **tailored prompts to support more fine-grained and targeted evaluation** unlike previous datasets which mostly rely on a fixed prompt. e.g., we design a specialized prompt to compare LaTeX formulas.
> >
> > - Existing studies also suggest that, for QA tasks, LLM-based evaluations are more aligned with human judgments than metrics such as accuracy or F1 [1,2,3].
> >
> > - We also conduct a consistency study between LLM-based evaluations and human judgments. Analysis on 66 examples shows that LLM-based and human evaluations are largely consistent, with an agreement rate of approximately 83%.
> >
> >     |LLM\Human|0|1|
> >     |---|---|---|
> >     |0|25|7|
> >     |1|4|30|
> >
> > 2. **Efficiency**
> >
> > In addition to higher reliability, our subjective evaluation functions improve efficiency through output reformatting (line 176). Unlike prior LLM-based evaluations that rely on long-form free-text comparison, our approach requires models to output only the key information, reducing both time and cost.
> >
> > Following the reviewer’s suggestion, we provide an empirical estimate: evaluating all 1,246 examples takes approximately 35 minutes and costs $0.056. This process can be further accelerated using simple parallelization techniques if needed.
> >
> > **References**
> >
> > [1] Wang, Cunxiang, et al. “Evaluating Open-QA Evaluation.” Advances in Neural Information Processing Systems, vol. 36, Curran Associates, Inc., 2023, pp. 77013–77042.
> >
> > [2] Ho, Xanh, et al. "LLM-as-a-Judge: Reassessing the Performance of LLMs in Extractive QA." arXiv preprint arXiv:2504.11972 (2025).
> >
> > [3] Kamalloo, Ehsan, et al. "Towards Robust QA Evaluation via Open LLMs." Proceedings of the 47th International ACM SIGIR Conference on Research and Development in Information Retrieval. ACM, 2024, pp. 2811–2816.

---

### Official Review · Reviewer_msPL · 2025-11-02

**Soundness:** 3
**Presentation:** 3
**Contribution:** 2
**Rating:** 4
**Confidence:** 4

**Summary:**

The paper introduces M4PQA, which is a framework for AI research question-answering. The benchmark is a human-annotated Multi-Modal Multi-Task Multi-Paper Question Answering dataset. The authors recruited 26 students and collected 1,246 examples and 13,948 papers in AI and created a set of questions with different types (single, double, etc.) and element categories (text, image, table, etc).

**Strengths:**

M4PQA targets research QA and retrieval rather than generic QA. The function-based, instance-level evaluation is a sensible move away from free-form semantic metrics and reduces grading ambiguity. The agentic baselines and the released action space/environment help standardize comparisons for tool-using agents, and the dataset statistics (balance across question types; mix of elements) suggest measured coverage.

**Weaknesses:**

Evaluation relies substantially on synthetically generated content/trajectories. The paper does not describe where human work ends and LLM synthesis begins. The “26 students” section says annotators read papers, pose questions, and wrap evaluators, but elsewhere EXTRACTOR agents generate QAs and trajectories; it remains unclear, per item type, whether humans created or only vetted the questions and how often LLM-authored items passed review. Labeling is described as function-based with optional subjective paths; however, the paper should state which splits or item types invoke an LLM judge.

Reported performance also appears tightly coupled to tool access (search capabilities, for example). Table 4 shows large jumps when agents can query the environment; that’s informative, but it confounds “reasoning” with “tool use.”

Qwen fine-tuning with EXTRACTOR trajectories predictably improves performance of small models, but it doesn’t clarify whether gains come from better tool formats, better retrieval behavior, or overfitting to the dataset’s answer-format conventions; no cross-family or cross-benchmark transfer is shown.

The dataset appears calibrated to be hard, which is fine, but the paper implicitly values difficulty without probing whether item construction systematically reduces performance on weaker models and favors stronger ones (e.g., formatting-sensitive outputs).

**Questions:**

No particular question at this time.

**Details Of Ethics Concerns:**

The paper uses students to label and collect data, but it is not clear whether this is approved by the institutional IRB.

---

> ### Author Response · Authors · 2025-11-25
> **Rebuttal to Reviewer msPL (1/3)**
>
> We would like to sincerely thank the reviewer for your kindness, valuable advice, and precious time in reviewing our work.
>
> **Q1: Contributions of this Paper**
>
> **A1:** Thank you for raising this point. To clarify, **M4PQA** and **ExTrActor** serve distinct purposes in our work: M4PQA provides human-annotated evaluation data, while ExTrActor generates synthetic training data.
>
> 1. M4PQA is a fully human-annotated dataset, **with no LLM involvement**, serving as an **evaluation set**. It corresponds to Section 2 and also the “26 students” section.
> 2. ExTrActor is a LLM-based framework, **without human intervention**, that generates **training data**, including QA examples and agentic trajectories, for fine-tuning agentic frameworks. It corresponds to Section 3.
>
> - We do not evaluate any method on ExTrActor-generated examples; they are used only to produce trajectories for training. **All evaluations are conducted on M4PQA** to provide a straightforward comparison between fine-tuned models and baseline models with different methods.
>
> - Regarding the reliability of ExTrActor-generated examples, results on M4PQA (Table 6) already show that these synthetic examples help improve performance without human annotations, suggesting their adequacy for fine-tuning. But we agree that further analysis of LLM-generated examples could strengthen this claim.
>
> - We sampled 120 synthetic examples and evaluated them with DeepSeek-V3.1 along three aspects: **format**, **correctness**, and **difficulty**, obtaining average scores of 4.5, 3.2, and 2.6 out of 5, respectively. A manual check on a subset further shows that its judgments are largely consistent with human assessments, indicating that although these examples are not very challenging, they remain reliable for fine-tuning.
>
> **Q2: Implementation Details of Instance-Level Evaluation**
>
> **A2:** We would like to resolve your concerns from the following two perspectives:
>
> 1. **Use of LLM Judge**
>
> - We mention in the title that M4PQA applies instance-level evaluation, which means that for each example, its annotator should **clarify and record the evaluation function and arguments used along with the question** in the following format (also shown in Appendix A.3):
>
>     ```json
>     {
>         "example": {
>             "eval_func": "eval_string_exact_match",
>             "eval_kwargs": {
>                 "gold": "Italian",
>                 "lowercase": true
>             }
>         }
>     }
>     ```
>
>     In fact, the use of LLM judge **depends on the question itself rather than on splits or element categories**. When an example requires complex semantic understanding beyond simple matching, the annotator may select a subjective function that invokes LLM calls. This instance-level evaluation enhances flexibility and enables more accurate assessment of factual correctness.
>
> - On the other hand, only 29% of evaluations require LLM judges. And empirically, evaluating all 1,246 examples takes approximately 35 minutes and costs $0.056, and this can be further accelerated using simple parallelization techniques.
>
> 2. **Adoption of output reformatting**
>
> - As discussed in Section 2.2 (line 178), output reformatting is employed to guide LLMs to generate the scoring points of primary interest, facilitating subsequent evaluation. While output reformatting is widely used in general [1, 2, 3, 4], its application within the QA domain has been little explored.
>
> - Our empirical check on 30 randomly sampled answers from Qwen2.5-3B-Instruct further shows that the model adheres to the answer format in over 90% of cases (28/30), indicating that output reformatting is quite applicable in practice.

---

> > ### Author Response · Authors · 2025-11-25
> > **Rebuttal to Reviewer msPL (2/3)**
> >
> > **Q3: Improvement Analysis for ExTrActor**
> >
> > **A3:** We appreciate the reviewer’s suggestion that improvements should be properly attributed and the framework’s generalization discussed; we address these points separately.
> >
> > 1. **Sources of Gains**
> >
> > We sample 10 examples for each question type where fine-tuned Qwen2.5-7B-Instruct outperform untrained baseline, totaling 40 examples. For each example, we analyze the factors contributing to improvement. Following the reviewer’s advice, we classify the improvements into three key contributors:
> >
> > - **Better Retrieval Strategy**: The model develops a clearer understanding of the overall retrieval process and the specific actions required at each step, allowing it to **plan more effectively** and identify useful intermediate steps.
> > - **Better Retrieval Behavior**: The model interacts more accurately with the environment, correctly interpreting schemas and providing appropriate tool parameters.
> > - **Better Understanding and Reasoning**: The model demonstrates deeper comprehension of the question and context, producing more coherent and reliable reasoning.
> >
> > |               | single | multiple | retrieval | comprehensive | Total |
> > | ------------- | ------ | -------- | --------- | ------------- | ----- |
> > | Strategy      | 4      | 7        | 1         | 1             | 13    |
> > | Behavior      | 3      | 1        | 4         | 6             | 14    |
> > | Understanding | 3      | 2        | 5         | 3             | 13    |
> > | Total         | 10     | 10       | 10        | 10            | 40    |
> >
> > The analysis shows that the dominant contributor varies across question types. While improved retrieval behavior is not always the primary factor, **reductions in tool usage errors are consistently observed**, enhancing both retrieval behavior and strategy as the model learns to use tools more effectively. This finding aligns with the component ablation in Section 4.3 and reinforces the claim that **our training framework enhances the model’s ability to generate valid actions**.
> >
> > 2. **Generalization of the Method**
> >
> > | SciDQA | table | image | from. | AVG |
> > |---|---|---|---|---|
> > | Qwen2.5-3B-Instruct | 35.6 | 37.0 | 33.9 | 35.2 |
> > | Qwen2.5-3B-Instruct w/ SFT | 44.8 | 44.5 | 38.8 | 43.3|
> >
> > | M4PQA | sgl. |multi. | retr. | comp. | AVG |
> > |---|---|---|---|---|---|
> > | Llama3.1-8B-Instruct | 10.3 | 1.9 | 13.5 | 8.1 | 8.4 |
> > | Llama3.1-8B-Instruct w/ SFT | 25.9 | 5.0 | 39.6 | 14.1 | 21.0 |
> >
> > We evaluate our model on SciDQA [5], another QA dataset on scientific papers, and fine-tune Llama3.1-8B-Instruct on the same training data generated by ExTrActor. Evaluation results indicate that our method consistently improves performance across different model families and datasets.
> >
> > **Q4: Baseline Selection**
> >
> > **A4**: In designing this dataset, we selected papers from the past two years to minimize the possibility of models recalling stored facts. As reflected in the evaluation results, agent performance shows substantial improvement when allowed to query the environment, indicating that external knowledge beyond the model’s internal parameters is crucial for answering these questions accurately.
> >
> > Regarding the comment that we “confound ‘reasoning’ with ‘tool use’”, we carefully reviewed the manuscript and did not identify such a confounding. We would appreciate further clarification from the reviewer if a specific instance is found.
> >
> > **Q5: Ethics Concerns**
> >
> > **A5**: We thank the reviewer for pointing out potential ethical concerns. Prior to starting the annotation project, we consulted our institution’s IRB, which confirmed that formal review was not required. Due to double-blind policy, we are unable to provide the full text of the ethical review guidelines; however, we confirm that all participating students are at least 18 years old, and no personal or private information was collected or used during the process.

---

> > ### Author Response · Authors · 2025-11-25
> > **Rebuttal to Reviewer msPL (3/3)**
> >
> > **References**
> >
> > [1] Liu, Michael Xieyang, et al. “We Need Structured Output: Towards User-centered Constraints on Large Language Model Output” Extended Abstracts of the CHI Conference on Human Factors in Computing Systems. ACM, 2024, pp. 1–9.
> >
> > [2] Shen, Zhengyuan, et al. “SLOT: Structuring the Output of Large Language Models.” Proceedings of the 2025 Conference on Empirical Methods in Natural Language Processing: Industry Track. Association for Computational Linguistics, 2025, pp. 472–491.
> >
> > [3] Hu, Xueyu, et al. “InfiAgent-DABench: Evaluating Agents on Data Analysis Tasks.” Proceedings of the 41st International Conference on Machine Learning (ICML’24). JMLR.org, 2024, pp. 1–29.
> >
> > [4] Shorten, Connor, et al. “StructuredRAG: JSON Response Formatting with Large Language Models” arXiv preprint arXiv:2408.11061.
> >
> > [5] Singh, Shruti, et al. “SciDQA: A Deep Reading Comprehension Dataset over Scientific Papers.” Proceedings of the 2024 Conference on Empirical Methods in Natural Language Processing. Association for Computational Linguistics, 2024, pp. 20908–20923.

---

### Official Review · Reviewer_KUWD · 2025-11-05

**Soundness:** 4
**Presentation:** 3
**Contribution:** 3
**Rating:** 6
**Confidence:** 4

**Summary:**

This paper introduce lots work to
0) frame question answering as multi turn agentic calling
1) create enviornments to generate agentic tractergories recognize the limits of human-input
2) showcase scaling law to small 7b/14B model

**Strengths:**

There are novelty associated with this approach, constructing tools like vector search, SQL. The questions themselves

There are a human baseline that helps with establishing a good baselines. this is a really important baseline

The paper also mentioned error-analysis of gpt-4o, and classifier them into different classes, i think the lack of context and over confidence are really important to address in future work.

**Weaknesses:**

The multi-turn setup might be more ideal for RL instead of SFT, but I recognized most of the work are more on engineering the multi agent prompts, setting up the right environment, and reading through raw responses.

**Questions:**

Why do you only compute loss on the model's last output? this means you are not doing any supervision to the actual tractegory

For the human baseline, does the error distrubtion is super similar to the gpt-4o error on this benchmark?

---

> ### Author Response · Authors · 2025-11-25
> **Rebuttal to Reviewer KUWD**
>
> We would like to sincerely thank the reviewer for your kindness, valuable advice, and precious time in reviewing our work.
>
> **Q1: RL-based Search Agents and Our Contributions**
>
> **A1:** We understand and acknowledge your opinion that LRM represents a current research trend, especially on multi-turn tasks. While part of our work indeed involves implementing and evaluating agentic baselines, we would like to reiterate the other two central contributions of our paper:
>
> 1. M4PQA, a new **fully human-annotated** dataset for multi-task multi-paper question answering, which requires extensive manual effort.
> 2. ExTrActor, a **fully human-free** framework for generating training data in the form of agentic trajectories.
>
> In parallel, a member of our team is currently exploring this area by **enhancing models’ multi-step reasoning capabilities through reinforcement learning**. We aim to refine this work further, so detailed and complete experimental results are not included in this paper. We provide below a subset of preliminary results obtained using GRPO:
>
> |Method|sgl.|multi.|retr.|comp.|AVG|
> |---|---|---|---|---|---|
> |SFT|21.08|4.95|51.04|22.18|24.07|
> |SFT + GRPO|22.22|6.19|52.78|20.42|24.24|
>
> Training is conducted on Qwen2.5-7B-Instruct, and results demonstrate that reinforcement learning yields further performance improvement within just 300 training steps. We will continue to advance our work in this direction in future research.
>
> **Q2: Implementation Details of ExTrActor**
>
> **A2:** Thank you for raising the question about why we compute the loss only on the model’s last output. Following the notation in line 333, the full trajectory generated by Actor is represented as $(u_0, a_0, \dots, u_i, a_i, \dots, u_n, a_n)$. Applying the sliding window yields **a set of contiguous segments**:
>
> $$
> \\{\\, (u_s,a_s,\dots,u_t,a_t) \mid 0 \le t \le n,\\; s = \max(0,\\, t-k+1) \\,\\}
> $$
>
> This converts a full trajectory into multiple partial trajectories. For each partial trajectory, we mask the earlier turns and compute the loss only on the final turn. Every intermediate step appears exactly once as the last turn of a partial trajectory (except those later removed during error removal), **ensuring full coverage of the entire interaction**.
>
> **Q3: Human Baseline Analysis**
>
> **A3:** The human and LLM baselines operate under different conditions. Human participants are limited to open-web search, whereas LLM-based agents can directly access curated databases and vectorstores, which offer more reliable retrieval.
>
> Under the human baseline setting, participants are not required to record intermediate reasoning or search steps, making example-level error analysis infeasible. However, we do collect their overall impressions after completing the test. All three participants report difficulty in identifying the correct papers, particularly when multiple sources are involved, and two additionally note challenges in understanding domain-specific terminology.
>
> The evaluation results below (Table 5 and 12) further show that although humans consistently outperform GPT-4o across question types and element categories, the margin on retrieval questions is relatively small. This indicates that the performance bottlenecks for humans differ from those of LLMs.
>
> | Baseline | sgl.  | multi. | retr. | comp. | text  | table | image | form. | meta. | AVG   |
> | -------- | ----- | ------ | ----- | ----- | ----- | ----- | ----- | ----- | ----- | ----- |
> | GPT-4o   | 45.58 | 10.53  | 52.13 | 35.56 | 39.61 | 23.00 | 25.60 | 33.86 | 47.54 | 35.96 |
> | Human    | 64.29 | 54.00  | 52.17 | 56.82 | 53.12 | 56.07 | 67.05 | 50.00 | 55.00 | 56.63 |

---

### Author Response · Authors · 2025-12-03
**General Reply to All Reviewers and Area Chairs**

We would like to thank all reviewers and area chairs for your patience and precious time, as well as many constructive and insightful suggestions which help us further polish our work.

Our paper introduces **M4PQA**, a comprehensive QA dataset for AI Research with instance-level evaluation, and **ExTrActor**, an automated framework for instruction data synthesis. These two components serve distinct purposes: M4PQA provides reliable, human-annotated evaluation data, whereas ExTrActor produces training data without human intervention. We also present extensive experiments and analyses to validate the quality of M4PQA and the effectiveness of ExTrActor.

We are therefore grateful and encouraged that our work is considered “**novelty associated**” and “**showcase scaling law**” (Reviewer KUWD), “**dataset statistics suggest measured coverage**” (Reviewer msPL), “**addresses a well-motivated and relevant problem**” (Reviewer b9vj), “**address shortcomings of other scientific papers benchmarks**” and “**highlighting the scalability of the framework**” (Reviewer ukJ4). We are also delighted that all reviewers rated our work from good to excellent in terms of soundness and presentation.

To address the reviewers’ questions, we conducted the following supplementary experiments and analyses:
- **E1. Cross-family, cross-dataset experiment for ExTrActor.** Corresponding to Q3 for Reviewer msPL.
- **E2. Comparison with other question-generation methods.** Corresponding to Q1 for Reviewer b9vj.
- **E3. Preliminary exploration of applying GRPO.** Corresponding to Q1 for Reviewer KUWD and Q2 for Reviewer ukJ4.
- **A1. Improvement analysis of ExTrActor.** Corresponding to Q3 for Reviewer msPL.
- **A2. Performance analysis of the human baseline.** Corresponding to Q3 for Reviewer KUWD.
- **A3. Quality analysis of examples and trajectories generated by ExTrActor.** Corresponding to Q1 for Reviewer msPL, b9vj and Q2 for Reviewer ukJ4.
- **A4. Efficiency analysis of ExTrActor.** Corresponding to Q1 for Reviewer b9vj.
- **A5. Efficiency analysis of M4PQA evaluation.** Corresponding to Q2 for Reviewer msPL and b9vj.

We would also like to remind the area chair that:
- During the discussion period, Reviewer ukJ4 stated on Nov 26 that we had addressed the main concern and that **the score would be raised to 6**.
- Though we were unable to receive further feedback from the other three reviewers due to the circumstances of this year’s review process, we believe that the potential misunderstanding and uncertainty could be reasonably resolved within the content of the paper and the clarifications provided in our rebuttal.

Once again, we sincerely thank all reviewers and area chairs for your valuable and professional contributions towards enhancing our manuscript, especially at a time when the peer-review ecosystem is facing unprecedented strain. We have uploaded a revised version with key changes clearly highlighted. The indices referenced in the rebuttal correspond to those in the original manuscript.

Thank you for your thorough and responsible reviews!

---

### Meta-Review · Area_Chair_gnG6 · 2026-01-07

**Summary:**

The authors proposed M4PQA as a human-annotated benchmark for scientific question answering. They also developed the EXTRACTOR framework to automate agent trajectory synthesis for instruction tuning. Reviewers raised concerns about the reliability of the synthetic training data and the choice of SFT over RL. Other issues include potential data contamination and the novelty of the benchmark.
The rebuttal resolved these points with manual quality scores and a detailed overlap analysis. The authors provided preliminary GRPO results to show improvements from RL. They also performed a contrastive analysis to show how M4PQA differs from previous datasets. Yet, a difficulty analysis (from Reviewer msPL) for smaller models is still missing from the work.  Overall, the paper is technically solid and provides a valuable contribution, so AC recommends acceptance.

**Reviewer Concerns:**

One concern appears not having been addressed by authors. "The dataset appears calibrated to be hard, which is fine, but the paper implicitly values difficulty without probing whether item construction systematically reduces performance on weaker models and favors stronger ones (e.g., formatting-sensitive outputs)." Authors should address this question in their final version.

**Reviewer Scores:**

Original scores were 6,4,6,4. The last reviewer mentioned they would increase the rating. AC thinks that the second reviewer might've increased the rating, too.

---

### Decision · Program_Chairs · 2026-01-26

Accept (Poster)